# RLIE: Rule Generation with Logistic Regression, Iterative Refinement, and Evaluation for Large Language Models

**Yang Yang** [* 1]  **Hua XU** [* 1]  **Zhangyi Hu** [* 1]  **Yutao Yue** [1 2]

## Abstract

Large Language Models (LLMs) can propose natural-language rules, circumventing the reliance on a predefined predicate space in traditional rule learning. However, existing LLM-based methods often neglect the global interactions among rules, and the potential of using fine-grained rule importance scores to calibrate neuro-symbolic reasoning remains underexplored. To address this gap, we introduce **RLIE**, a framework that integrates LLMs with probabilistic modeling to learn weighted rule sets in four stages: (1) **R**ule generation: proposing and filtering candidate rules via LLMs; (2) **L**ogistic regression: learning sparse, calibrated weights for global rule selection; (3) **I**terative refinement: revising the rule set with error-driven hard examples; and (4) **E**valuation: validating the learned system via comparative inference paradigms. Across multiple real-world datasets and LLM backbones, our learned weighted rules achieve superior stability and accuracy, whereas rule-injection prompting yields mixed results and often degrades performance. These results suggest LLMs excel at semantic rule discovery but are less reliable at controlled probabilistic aggregation. Our findings highlight both the promise and the limits of LLMs for inductive reasoning, motivating a principled integration with classic probabilistic rule combination for reliable neuro-symbolic reasoning.[1]

[1]The Hong Kong University of Science and Technology (Guangzhou), Guangzhou 511400, China [2]Institute of Deep Perception Technology, JITRI, Wuxi 214000, China. Correspondence to: Yutao Yue <yutaoyue@hkust-gz.edu.cn>.

*Proceedings of the 43$^{rd}$ International Conference on Machine Learning*, Seoul, South Korea. PMLR 306, 2026. Copyright 2026 by the author(s).

## 1. Introduction

In many data-driven applications and scientific inquiry, the goal is increasingly shifting from pure prediction to constructing verifiable, reusable, and composable theories (Zhou et al., 2024; Yang et al., 2024a; Minh et al., 2022). Such theories enable explainable and auditable decisions and, in scientific contexts, can serve as building blocks for discovering underlying structures and accumulating knowledge (Yang et al., 2023; 2024b). These theories may be expressed as formal, structural statements (Cohen et al., 1995; Cropper & Morel, 2021) or as natural-language hypotheses[2] (Zhou et al., 2024); regardless of form, they are declarative and testable units whose predictions can be verified by external evidence. A longstanding challenge is how to learn a collection of such testable hypotheses and integrate them into a coherent rule set that captures regularities, supports reliable decisions, and enables cumulative scientific understanding (Bazgir et al., 2025).

To ground this discussion, consider spam detection as a binary classification task. A predicate-based rule set might include three predicate statements:

1. $\text{HasToken}(\texttt{prize}, x) \wedge \text{HasFile}(x) \rightarrow \text{Spam}(x)$;

2. $\text{InBlacklist}(x) \wedge \text{HastLink}(x) \rightarrow \text{Spam}(x)$;

3. $\text{TitleAllCaps}(x) \wedge \text{HasToken}(\texttt{free}, x) \rightarrow \text{Spam}(x)$.

A deterministic inference framework is to predict spam if any of the conditions is satisfied (Fürnkranz & Kliegr, 2015). Such classical rule learning methods depend on a predefined predicate space with limited expressiveness for unstructured inputs and open-ended semantics (Cerna & Cropper, 2024; Hocquette et al., 2024). In contrast, probabilistic approaches treat each rule's firing as an indicator ($r_j \in \{0, 1\}$) and learn global weights – e.g., via logistic regression – to combine them into a unified prediction (Ruczinski et al., 2003; Friedman & Popescu, 2008). Crucially, this probabilistic formulation allows for greater flexability, accommodating rules defined as either symbolic predicates or neural indicators.

The advent of large language models (LLMs) opens a new

[2]In this paper, we use "rule" and "hypothesis" interchangeably, and refer to both as "rule".

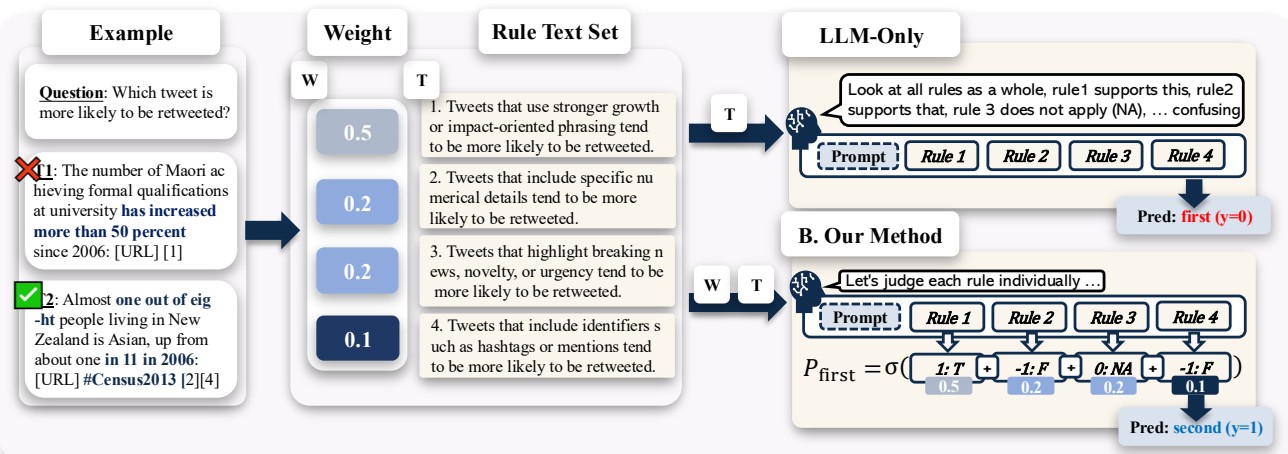

*Figure 1.* **Benefit of explicit aggregation.** LLM-only prompting reasons over conflicting rule signals, whereas RLIE separates rule-wise judging from global aggregation and uses a weighted logistic combiner to integrate evidence and resolve conflicts for reliable prediction.

opportunity for learning expressive rules in natural languages: given a few examples and task context, LLMs can generate rules directly in natural language (Ellis, 2023; Singh et al., 2022). Such rules can articulate fine-grained conditions, relations, and exceptions that are cumbersome to enumerate with a fixed predicate space. However, existing LLM-based methods typically either iteratively optimize a single rule (Qiu et al., 2023) or generate multiple rules without considering their global interdependence. This often leads to misalignment between training and inference (Yang et al., 2023; Zhou et al., 2024; Yang et al., 2024b), leaving the challenge of learning a compact and coordinated rule set unresolved. More broadly, the probabilistic composition of natural-language rules remains under-studied: it is unclear whether LLMs can robustly apply rule weights for reliable aggregation. Motivated by our empirical findings on the limitations of LLM probabilistic reasoning (see Section 5.2), we propose a framework that combines the expressive power of LLMs with the calibrated composition of classical probabilistic models. See Figure 1 for an illustrative example.

To realize this synergy, we propose **RLIE**, a unified framework integrating **R**ule generation, **L**ogistic regression, **I**terative refinement, and **E**valuation. RLIE first prompts an LLM to generate a pool of candidate rules. To enable learning, it uses the LLM to produce instance-level judgments for each rule. A regularized logistic regression model is then fitted to learn probabilistic weights, acting as explicit importance scores for rule selection. RLIE iteratively improves the rule set by mining hard examples to guide the LLM in revising or extending rules. Finally, the Evaluation stage verifies the system by comparing the learned logistic combiner against prompt-based inference. Our contributions can be summarized as follow:

**1.** We advocate a principled *division of labor*: LLMs handle semantic rule induction and local assessment, while statistical models provide globally calibrated composition, balancing semantic flexibility with inference robustness.

**2.** We propose a hard-example-driven refinement mechanism that improves the rule set through reflective revision and targeted extension, promoting rule quality and diversity.

**3.** We evaluate inference paradigms, finding that although LLMs excel at proposing interpretable rules, they struggle to effectively utilize rule weights for precise reasoning. This highlights a key deficiency in LLM reasoning and underscores the necessity of an external probabilistic combiner.

## 2. Related Work

### 2.1. Classical Rule Learning and Its Applications

**Rule Learning**. Classical approaches encompass Inductive Logic Programming (ILP) (Cropper et al., 2022), association rule mining, and differentiable neuro-symbolic methods (Qiao et al., 2021; Glanois et al., 2022; Yang et al., 2024a). Despite their methodological differences, these paradigms share a common objective: inducing human-readable, reusable discriminative rules within a structured hypothesis space. A fundamental limitation remains their reliance on a predefined predicate vocabulary, which restricts expressiveness and hinders applicability to unstructured inputs with open-ended semantics.

**Rule Organization and Reasoning**. Inference semantics dictate how multiple rules interact to form a decision. One stream of research utilizes ordered rule lists (e.g., decision lists) to encode priority, exceptions, and defaults via sequential matching (Yang et al., 2017; Xu et al., 2024). Conversely, unordered rule sets integrate evidence in parallel, enabling global trade-offs and model compression (Qiao et al., 2021; Yang & van Leeuwen, 2022). Within unordered settings, deterministic aggregators (e.g., OR/AND,

thresholding, voting) offer simplicity but often prove brittle under conflicts, coverage gaps, and inter-rule dependencies. Probabilistic aggregation mitigates these issues by modeling uncertainty and improving calibration. A prominent interpretable instantiation is the linear log-odds model, which treats rule satisfaction $r_j(x) \in \{0, 1\}$ as a feature and learns a weighted combination (Ruczinski et al., 2003)

$$\Pr(y = 1 \mid x) = \sigma\left(\beta_0 + \sum_j \beta_j r_j(x)\right).$$

Our work builds upon this foundation but departs from the standard assumption that rules exist in a fixed, executable symbolic form with directly computable satisfactions. Instead, we focus on natural-language rules, whose instance-level satisfactions must be judged (and may involve abstention) by LLMs. This distinction motivates a principled division of labor: employing LLMs for *local* rule interpretation and instance-level judgments, while retaining a logistic combiner for *global* selection and calibrated aggregation. The resulting hybrid inference preserves the semantic expressiveness of natural-language rules while ensuring the stability and interpretability of the overall decision process.

### 2.2. LLM-based Rule Learning and Its Applications

**Rule Learning**. Current LLM-based induction falls into two paradigms: *single-hypothesis refinement* (Qiu et al., 2023), which iteratively optimizes one rule via feedback, and *multi-hypothesis* methods like HypoGeniC (Zhou et al., 2024) that maintain and update a candidate pool. Closely related, Zhong et al. (Zhong et al., 2024) learn natural-language predicates as binary features and compose them with a statistical readout. However, these methods either treat hypotheses in isolation or operate on predicate-level features with relatively simple refinement, reflecting a trade-off between intensive local refinement, diverse hypothesis retention, and explicit rule interaction modeling.

**Rule Organization and Reasoning**. For rule-based LLM reasoning, existing work often uses the LLM as the inference engine via rule-augmented prompting (Zhang et al., 2024; Zhou et al., 2024). While effective in specific settings, implicit prompt-based aggregation becomes unstable as rule sets expand or conflict. Moreover, this black-box approach hinders error attribution, calibration, and the analysis of rule interactions, motivating our move toward explicit probabilistic composition.

In contrast, our work investigates *explicit probabilistic composition* for adaptive natural-language predictive rules, rather than relying on either prompt-based aggregation or predicate-level readouts alone. We systematically compare direct probabilistic combination against prompt-based inference under progressively richer injected information (rules, weights, and prediction references), demonstrating the necessity of an external, calibrated aggregator.

## 3. Method

We consider a binary classification task on a labeled dataset $\mathcal{S} = \{(x_i, y_i)\}_{i=1}^N$, where $x_i \in \mathcal{X}$ is a natural-language text and $y_i \in \{0, 1\}$ is its label. We split $\mathcal{S}$ into training, validation, and test sets, denoted by $\mathcal{S}_{\text{train}}$, $\mathcal{S}_{\text{val}}$, and $\mathcal{S}_{\text{test}}$, respectively. Our goal is to learn a set of natural-language rules $\mathcal{H}^\star = \{h_1, \ldots, h_m\}$ together with aggregation weights $\theta$ that linearly combine these rules to model the discriminative relation between $x$ and $y$. We constrain the rule set size by $m \leq H$, where $H$ is a predefined capacity limit. Our framework consists of four stages (Figure 2):

**1. R**ule Generation: Starting from $\mathcal{H} := \emptyset$, we prompt an LLM with a randomly sampled subset of $\mathcal{S}_{\text{train}}$ to propose candidate rules, and filter them by coverage to obtain the initial rule set $\mathcal{H}^{(1)}$.

**2. L**ogistic Regression: Given $\mathcal{H}^{(t)}$, we train a logistic regression combiner on the whole training set $\mathcal{S}_{\text{train}}$ to learn rule weights $\theta^{(t)}$, with hyperparameters selected on $\mathcal{S}_{\text{val}}$.

**3. I**terative Refinement: We select hard examples from $\mathcal{S}_{\text{train}}$ with the highest prediction errors from the combiner, and feed them back to the LLM together with the current rule set $\mathcal{H}^{(t)}$ to generate new rules $\Delta\mathcal{H}^{(t)}$ to augment the rule set. We then merge the updated rules and enforce the capacity constraint: if $|\mathcal{H}^{(t)} \cup \Delta\mathcal{H}^{(t)}| > H$, we prune rules based on validation performance. The combiner is retrained to update $\theta^{(t+1)}$. We repeat this process with early stopping monitored on $\mathcal{S}_{\text{val}}$.

**4. E**valuation: We strictly evaluate the finalized rule set and weights on $\mathcal{S}_{\text{test}}$. This stage quantifies the system's reliability and compares the calibrated probabilistic inference (combiner) against uncalibrated LLM reasoning (prompting), verifying the effectiveness of weighted rule set.

Our prompts are provided in Appendix E.

### 3.1. Rule Generation

**Initialization and Candidate Generation**. We start with an empty rule set $\mathcal{H}^{(0)} := \emptyset$. In the first iteration, we uniformly sample $k$ examples from $\mathcal{S}_{\text{train}}$ and prompt an LLM to propose $h$ candidate natural-language rules ($h \leq H$), denoted by $\mathcal{H}_{\text{cand}}^{(1)} = \{h_1^{(1)}, \ldots, h_h^{(1)}\}$.

**Individual Rule Application**. For each candidate rule $h_j^{(1)}$ and sample $x_i$ from the whole training set, we query an LLM-based judge to produce a ternary judgment $z_{i,j}^{(1)} = \text{LLM}\left(x_i, h_j^{(1)}\right) \in \{-1, 0, +1\}$, where $+1$ and $-1$ indicate predictions for the positive and negative class, respectively, and $0$ denotes abstention when the LLM deems the rule inapplicable to sample $x_i$. This explicitly models rule coverage and mitigates spurious forced predictions, facilitating sparse and robust aggregation in the subsequent stage.

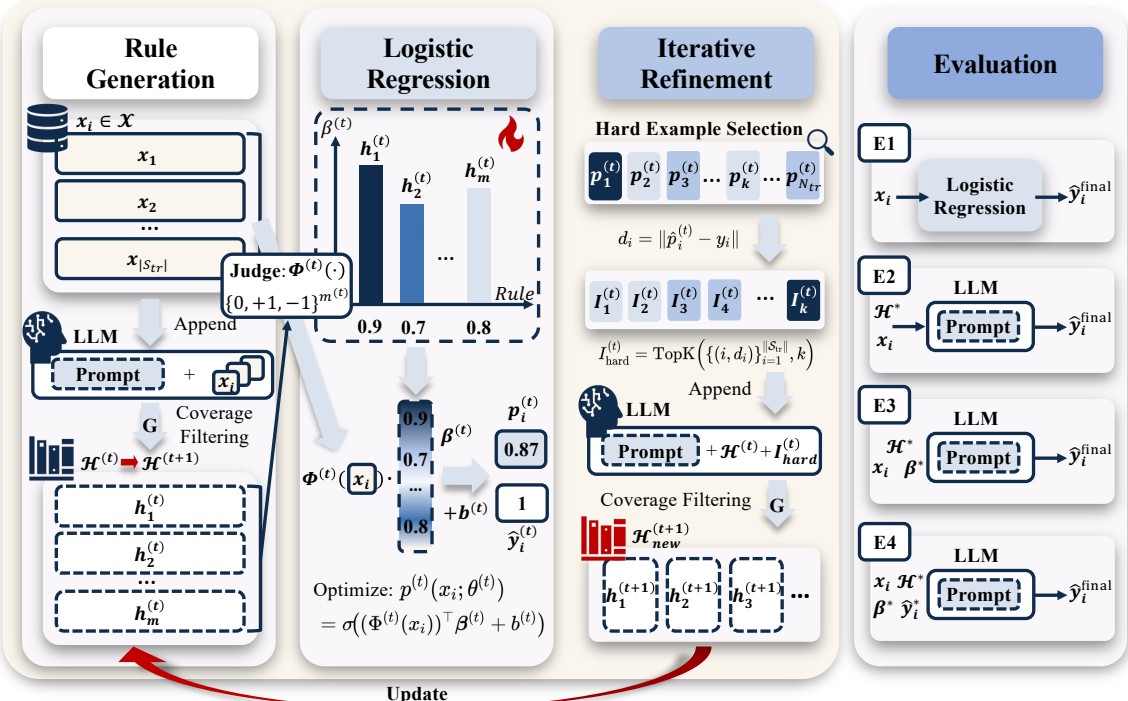

*Figure 2.* **Overview of RLIE.** RLIE first generates candidate natural-language rules, then learns their weights with logistic regression, refines the rule bank using hard examples, and evaluates the learned system under multiple inference protocols.

**Coverage-based Filtering and Rule Set Update**. We measure the coverage of each candidate rule $h_j^{(1)}$ on the training set as: $\text{Cov}_{\text{train}}\left(h_j^{(1)}\right) = \frac{1}{N_{\text{train}}} \sum_{i=1}^{N_{\text{train}}} \mathbb{I}\left[z_{i,j}^{(1)} \neq 0\right]$. We discard rules from $\mathcal{H}_{\text{cand}}^{(1)}$ with $\text{Cov}_{\text{train}}\left(h_j^{(1)}\right) < \gamma$ to ensure that the rule set is general and applicable to a significant portion of samples. We denote the remaining rules by $\mathcal{H}_{\text{new}}^{(1)}$. The rule set is then updated as $\mathcal{H}^{(1)} = \mathcal{H}^{(0)} \cup \mathcal{H}_{\text{new}}^{(1)}$.

### 3.2. Logistic Regression

Unlike neural networks which are sometimes overconfident(Guo et al., 2017), our logistic combiner minimizes log-loss, inherently promoting calibration.

**Feature Construction**. At iteration $t$, given a rule set $\mathcal{H}^{(t)} = \{h_1^{(t)}, \ldots, h_{m^{(t)}}^{(t)}\}$ where $m^{(t)} \leq H$, we define a mapping $\Phi^{(t)}$ from an input sample to a vector of rule application results $\Phi^{(t)}: \mathcal{X} \to \{-1, 0, 1\}^{m^{(t)}}$, $\Phi^{(t)}(x_i) = \mathbf{z}_i^{(t)} = [z_{i,j}^{(t)}]_{j=1}^{m^{(t)}}$.

**Weight Learning**. We model the positive-class probability using a regularized logistic regression combiner $p^{(t)}(x_i; \theta^{(t)}) = \sigma\left((\Phi^{(t)}(x_i))^\top \boldsymbol{\beta}^{(t)} + b^{(t)}\right)$ where $\sigma$ is the sigmoid function and $\theta^{(t)} = (\boldsymbol{\beta}^{(t)}, b^{(t)})$. We learn these parameters by minimizing the cross-entropy (CE) loss on $\mathcal{S}_{\text{train}}$ with Elastic Net (Zou & Hastie, 2005) regularization:

$$\boldsymbol{\beta}^{(t)}, b^{(t)} = \underset{\boldsymbol{\beta}^{(t)}, b^{(t)}}{\arg\min} \left[\frac{1}{|\mathcal{S}_{\text{train}}|} \sum_{x_i \in \mathcal{S}_{\text{train}}} \mathcal{L}\left(y_i, p^{(t)}(x_i; \theta^{(t)})\right) \right.$$
$$\left. + \lambda \left(\alpha \|\boldsymbol{\beta}^{(t)}\|_1 + \frac{1-\alpha}{2} \|\boldsymbol{\beta}^{(t)}\|_2^2\right)\right]$$

where $\mathcal{L}$ denotes binary cross-entropy. The $L_1$ term promotes sparsity (rule selection), while the $L_2$ term improves robustness. We select $(\lambda, \alpha)$ on $\mathcal{S}_{\text{val}}$ via stratified $K$-fold cross-validation, and refit on the full $\mathcal{S}_{\text{train}}$ to obtain $\hat{\theta}^{(t)}$.

**Label Prediction**. Given $\hat{\theta}^{(t)}$, we predict sample $x_i$ with $\hat{y}_i^{(t)} = \mathbb{I}\left[p^{(t)}(x_i; \hat{\theta}^{(t)}) \geq 0.5\right] \in \{0, 1\}$.

### 3.3. Iterative Refinement

Although the initial rule set $\mathcal{H}^{(1)}$ is derived from a random subset of $\mathcal{S}_{\text{train}}$, we subsequently refine it by targeting hard examples under the current model $(\mathcal{H}^{(t)}, \hat{\theta}^{(t)})$ for $t \geq 1$.

**1. Hard Example Selection**. For each $(x_i, y_i) \in \mathcal{S}_{\text{train}}$, we compute the predicted probability $\hat{p}_i^{(t)} = p^{(t)}(x_i; \hat{\theta}^{(t)})$ and define the prediction error $d_i = |\hat{p}_i^{(t)} - y_i|$. We then select the indices of the top-$k$ hardest examples: $\mathcal{I}_{\text{hard}}^{(t)} = \text{TopK}\left(\{d_i\}_{i=1}^{|\mathcal{S}_{\text{train}}|}, k\right)$.

**2. New Rule Generation**. We present the hard examples $\mathcal{I}_{\text{hard}}^{(t)}$ along with the current rule set $\mathcal{H}^{(t)}$ to the LLM,

prompting it to revise existing rules or propose $h$ new rules (see Figure 6). After applying the same coverage-based filtering as in Section 3.1, we obtain $\mathcal{H}_{\text{new}}^{(t+1)}$.

**3. Rule Set Update**. We merge new rules with the current set: $\mathcal{H}_{\text{tmp}}^{(t+1)} = \mathcal{H}^{(t)} \cup \mathcal{H}_{\text{new}}^{(t+1)}$. If $|\mathcal{H}_{\text{tmp}}^{(t+1)}| \leq H$, we set $\mathcal{H}^{(t+1)} = \mathcal{H}_{\text{tmp}}^{(t+1)}$. Otherwise, we prune rules by ranking them according to their F1 scores on $\mathcal{S}_{\text{val}}$ calculated on non-abstaining samples and retaining the top $H$ rules.

**4. Parameter Update and Termination**. Given $\mathcal{H}^{(t+1)}$, we refit the logistic regression combiner as described in Section 3.2 to obtain $\hat{\theta}^{(t+1)}$. The refinement terminates if validation performance fails to improve by at least $\delta$ for $p$ consecutive iterations, or if the maximum iteration count $R_{\max}$ is reached. We return $(\mathcal{H}^{\star}, \hat{\theta}^{\star})$ from the checkpoint with the best validation performance.

### 3.4. Evaluation

Upon obtaining the weighted rule sets, we employ various inference strategies to verify their quality and identify the optimal deployment protocol. These strategies fall into two paradigms: direct inference using the learned logistic combiner, and LLM-based inference where the prompt is progressively augmented with information from the learned system. Let $\text{LLM}(\cdot)$ denote an LLM that outputs the final label. We use $\mathbf{R}$ to denote the rule texts in $\mathcal{H}^{\star}$, $\mathbf{W}$ the learned weights $\hat{\theta}^{\star}$, and $\mathbf{P}$ the combiner's predicted label $\hat{y}_i^{\text{comb}} = \mathbb{I}\left[p(x_i; \hat{\theta}^{\star}) \geq 0.5\right]$.

**(E1) Combiner**. We set $\hat{y}_i^{\text{final}} = \hat{y}_i^{\text{comb}}$, directly utilizing the combiner's output as the final prediction without further LLM prompting at inference time.

**(E2) LLM(R)**. The LLM is provided with the input $x_i$ and rule texts $\mathbf{R}$: $\hat{y}_i^{\text{final}} = \text{LLM}(x_i, \mathbf{R})$. This tests the LLM's intrinsic ability to reason over a set of natural-language rules absent of explicit weight signals.

**(E3) LLM(RW)**. The LLM is provided with $x_i$, $\mathbf{R}$, and the learned weights $\mathbf{W}$: $\hat{y}_i^{\text{final}} = \text{LLM}(x_i, \mathbf{R}, \mathbf{W})$. This tests whether explicit probabilistic importance cues facilitate the LLM in prioritizing and aggregating rule evidence.

**(E4) LLM(RWP)**. The LLM is provided with $x_i$, $\mathbf{R}$, $\mathbf{W}$, and the combiner prediction $\mathbf{P}$: $\hat{y}_i^{\text{final}} = \text{LLM}(x_i, \mathbf{R}, \mathbf{W}, \mathbf{P})$. This tests whether the combiner's prediction can serve as a reference signal to stabilize LLM-based aggregation in challenging cases.

## 4. Experiment Setup

### 4.1. Dataset

We evaluate RLIE on six real-world tasks, including **Reviews** (deceptive review detection), **Dreaddit** (mental stress detection), **Headlines** (headline engagement prediction), **Citations** (paper citation impact prediction), **LLM Detect** (AI-generated text detection), and **Retweets** (pairwise retweet prediction). All tasks are formulated as binary classification over natural-language inputs (e.g. given a review, we need to evaluate whether it is deceptive or not). Detailed dataset descriptions are provided in Appendis A.3.

### 4.2. Baseline

We compare RLIE with representative LLM-based inference and rule-learning baselines: **(1) Zero-shot**: direct prediction from task description without labeled demonstrations or rules. **(2) Few-shot (ICL)**: standard in-context learning with a small set of labeled demonstrations sampled from the training split. **(3) Zero-shot Gen**: generate a pool of candidate rules from task description only, then select the single best rule on the validation set for test-time inference. **(4) Zhong et al.** (Zhong et al., 2024): use learned natural-language predicate denotations as interpretable features for a logistic-regression readout. **(5) IO Refinement** (Qiu et al., 2023): iteratively propose-select-refine a single rule using an error-driven refinement loop. **(6) HypoGeniC** (Zhou et al., 2024): maintain and expand a hypothesis library, and perform top-$k$ hypothesis selection for prediction with reward-driven updates.

### 4.3. Experimental Details

**LLM backbones.** To ensure a fair comparison for all methods that *generate* or *refine* rules, we use DEEPSEEK-V3.2 as the default backbone throughout the main experiments. To examine whether our rule-generation framework is model-agnostic, we additionally run RLIE with alternative backbones (QWEN3-NEXT-80B and QWEN3-235B) while keeping the rest of the pipeline unchanged.

**Protocol and metrics.** We use the original train/validation/test splits provided by HypoBench for all methods, and report mean Accuracy (ACC) and Macro-F1 on the test set over three runs with different random seeds.

**RLIE hyperparameters.** We set the rule capacity to $H = 10$. In each iteration, we sampled $k = 20$ hard examples and generated $h = 5$ new rules. We filter candidate rules by a minimum training coverage threshold $\gamma = 0.2$. During iterative refinement, we set the max iteration number $R_{\max}$ to be 20. The sampling temperature is set to be $1 \times 10^{-5}$ for stability and reproducibility. The implementations of IO Refinement and HypoGeniC follow the original papers. Further details are provided in Appendix A.

## 5. Results

In this section, we analyse the overall performance of RLIE, conducting experiments on different inference strategies to

*Table 1.* Overall performance comparison (Accuracy / Macro-F1, $\times 10^{-2}$) on full datasets. RLIE (Ours) uses the Linear-Only inference strategy. The best results among generalizable methods are **bolded**.

| Method | Backbone | Reviews | Dreaddit | Headlines | Citations | LLM Detect | Retweets |
|---|---|---|---|---|---|---|---|
| Zero-shot | DeepSeek-V3.2 | 53.9 / 42.3 | 64.7 / 60.0 | 61.2 / 59.6 | 62.5 / 50.0 | 82.4 / 82.0 | 63.5 / 62.6 |
| Few-shot (ICL) | DeepSeek-V3.2 | 65.3 / 65.1 | 63.8 / 58.5 | 62.0 / 61.8 | 60.4 / 49.8 | 80.4 / 79.6 | 57.7 / 53.9 |
| Zero-shot Gen | DeepSeek-V3.2 | 65.4 / 64.8 | 67.2 / 63.9 | 57.5 / 57.4 | 46.1 / 47.7 | 63.1 / 57.6 | 58.1 / 58.1 |
| Zhong et al. | DeepSeek-V3.2 | 66.2 / 65.9 | 75.4 / 75.0 | 48.2 / 44.1 | 58.1 / 54.7 | 84.9 / 84.6 | 50.0 / 52.1 |
| IO Refinement | DeepSeek-V3.2 | 65.9 / 65.4 | 78.5 / 78.1 | 62.0 / 61.1 | 54.2 / 51.0 | 83.6 / 83.4 | 57.1 / 56.1 |
| HypoGeniC | DeepSeek-V3.2 | 69.1 / 69.3 | 80.5 / 80.5 | 59.9 / 60.1 | 46.9 / 49.3 | 85.2 / 85.1 | 61.9 / 61.8 |
| **RLIE (Ours)** | Qwen3-Next-80B | 68.3 / 67.8 | 81.1 / 81.1 | 61.1 / 60.9 | 56.3 / 55.0 | 87.6 / 87.6 | 61.9 / 61.8 |
| **RLIE (Ours)** | Qwen3-235B | **71.5 / 71.4** | 79.9 / 79.9 | 60.6 / 60.4 | 61.5 / 60.6 | 88.3 / 88.3 | **66.5 / 66.5** |
| **RLIE (Ours)** | DeepSeek-V3.2 | 70.9 / 70.7 | **82.3 / 82.3** | **67.0 / 67.0** | **64.6 / 63.0** | **90.7 / 90.7** | 65.7 / 65.6 |

find best way to apply the learnt rules. We also provide computational cost analysis and two case studies.

## 5.1. Main Results

To enable a direct comparison with baselines, we evaluate RLIE using the *combiner* for final prediction throughout this section. The results are summarized in Table 1, showing that RLIE is strongly competitive across all tasks. We further evaluate RLIE with three different backbones and observe a consistent trend: stronger backbones typically yield a higher overall ceiling, while even with a smaller backbone RLIE remains highly competitive on multiple tasks, indicating robust adaptability across different model capacity regimes.

Comparing against interpretable rule-based baselines reveals a clear progression in capability. ZERO-SHOT GEN generates a set of candidate rules and selects the best single rule for inference, which is limited in expressiveness as it relies on a single hypothesis. IO REFINEMENT iteratively improves a single rule, typically leading to stable gains by enhancing rule quality. HYPOGENIC extends this to a set of multiple rules, which can provide additional benefits in some cases but does not consistently outperform single-rule methods, suggesting that multi-rule inference requires a more reliable global integration mechanism. Building on these observations, RLIE couples iterative rule expansion with weight learning and global aggregation: it progressively addresses uncovered hard cases while explicitly learning rule contributions and combining them at the global level, thereby leveraging the benefits of rule sets more reliably and achieving consistent improvements across most tasks.

We also consider direct prompting (ZERO-SHOT/FEW-SHOT) in which LLMs directly output judgments without producing explicit rules, thus offering weaker interpretability. Notably, FEW-SHOT does not necessarily improve over ZERO-SHOT, indicating that inducing implicit generalization from a handful of examples is challenging, often failing to capture stable patterns from insufficient samples.

## 5.2. How to Best Utilize Rules?

To determine the optimal inference strategy for the learned probabilistic rules, we compare the four protocols (E1–E4) defined in Section 3.4 across different LLM backbones. Intuitively, an LLM might synthesize the rule texts, the learned weights, and even the combiner's reference prediction to produce superior judgments; however, Table 2 shows that this does not yield consistent gains. Overall, E1 (COMBINER) is the most stable and consistently strong strategy: it achieves the best or near-best Accuracy and Macro-F1 in most settings, generally outperforming the three prompting-based LLM inference variants. This suggests that the rule set learned by RLIE is of high quality, and that an explicit probabilistic combiner can more reliably translate multi-rule evidence into final decisions.

In contrast, injecting more information into the prompt does not reliably improve performance. Among all prompting-based variants, the rule-only method (E2) is often the strongest; adding rule weights (E3) does not lead to consistent improvements, suggesting that LLMs struggle to reliably exploit numeric or probabilistic signals under prompting. Further providing the combiner prediction as a reference (E4) is also not a dependable upgrade: while it occasionally matches or slightly exceeds E1, its performance is highly variable, rendering it unsuitable as a default strategy. Overall, when the model is tasked with jointly integrating multiple rules, weights, and reference signals, its final decision becomes sensitive to prompt structure and potential information conflicts, leading to inconsistent behavior.

Taken together, our results support a robust "division of labor": employing LLMs primarily for proposing and refining rules, while relying on the combiner for global, weight-aware aggregation and final inference. This synergy preserves interpretability while delivering better performance.

We also assess the reliability of the confidence scores in Appendix C, and RLIE achieves significantly lower Expected Calibration Error compared to direct prompting in most

*Table 2.* Impact of inference strategies (Accuracy / Macro-F1, $\times 10^{-2}$). **E1** is the most consistently strong approach, while prompting-based variants **(E2–E4)** yield mixed results. The best Accuracy and Macro-F1 for each backbone and dataset are **bolded**, respectively.

| Backbone | Strategy | Reviews | Dreaddit | Headlines | Citations | LLM Detect | Retweets |
|---|---|---|---|---|---|---|---|
| DeepSeek V3.2 | (E1) Combiner | **70.9 / 70.7** | 82.3 / 82.3 | **67.0 / 67.0** | **64.6 / 63.0** | **90.7 / 90.7** | **65.7 / 65.6** |
| | (E2) Prompt(R) | 65.7 / 63.1 | 77.5 / 76.6 | 66.9 / 66.8 | 60.4 / 56.2 | 89.6 / 89.6 | 64.4 / 64.3 |
| | (E3) Prompt(RW) | 65.4 / 64.0 | 77.9 / 77.1 | 65.2 / 65.0 | 55.2 / 53.5 | 85.2 / 85.0 | 62.5 / 61.8 |
| | (E4) Prompt(RWP) | 69.5 / 68.6 | **82.4 / 82.4** | 66.8 / 66.8 | 57.3 / 55.9 | 89.3 / 89.3 | 64.7 / 64.6 |
| Qwen3 235B | (E1) Combiner | **71.5 / 71.4** | **79.9 / 79.9** | **60.6 / 60.4** | **61.5 / 60.6** | 88.3 / 88.3 | **66.5 / 66.5** |
| | (E2) Prompt(R) | 64.5 / 64.2 | 77.3 / 76.7 | 60.0 / 59.7 | 60.4 / 60.3 | **88.4 / 88.4** | 64.5 / 63.2 |
| | (E3) Prompt(RW) | 63.4 / 63.2 | 77.7 / 77.1 | 58.5 / 58.2 | 60.4 / 60.1 | 87.1 / 87.1 | 65.8 / 65.0 |
| | (E4) Prompt(RWP) | 66.8 / 66.6 | 79.7 / 79.5 | 59.9 / 59.8 | 54.2 / 54.0 | 87.9 / 87.9 | 66.0 / 65.7 |
| Qwen3-Next 80B | (E1) Combiner | **68.3 / 67.8** | **81.1 / 81.1** | 62.9 / 62.7 | **56.3 / 55.0** | **87.6 / 87.6** | **61.9 / 61.8** |
| | (E2) Prompt(R) | 66.9 / 66.6 | 74.5 / 73.1 | 62.6 / 62.3 | **56.3** / 53.0 | 84.1 / 84.0 | 57.9 / 54.7 |
| | (E3) Prompt(RW) | 64.7 / 63.9 | 76.2 / 75.2 | **64.1 / 64.0** | 55.2 / 53.2 | 75.9 / 74.5 | 57.1 / 53.0 |
| | (E4) Prompt(RWP) | 67.7 / 67.4 | 79.5 / 79.1 | 62.0 / 61.9 | 55.2 / 53.7 | **87.6 / 87.6** | 58.9 / 56.4 |

cases, which confirms that RLIE effectively mitigates the overconfidence issue often observed in LLMs.

## 5.3. Computational Cost Analysis

We estimate the computational cost of the entire rule learning pipeline based on token usage statistics derived from our experiments (see Table 3). Compared to the robust baseline HYPOGENIC, RLIE reduces token consumption by approximately 8%, demonstrating that our hard-example driven refinement is more sample-efficient than maintaining a large rule bank. Although RLIE consumes more computational resources than the lightweight IO REFINEMENT, this additional cost is justified by our rigorous validation protocol (evaluating rules on the full validation set vs. only 10 samples in IO Refinement), which is crucial for ensuring the reliability of the learned neuro-symbolic system.

*Table 3.* Average computational cost per task. RLIE achieves a balance between efficiency and robustness, consuming fewer tokens than HypoGeniC while performing comprehensive validation. Energy costs are estimated by assuming a conservative energy intensity of 1.0 J/token for the MoE backbone.

| Method | Avg. Tokens (M) | Est. Energy (kWh) | Relative Cost |
|---|---|---|---|
| HypoGeniC | 13.24 | 3.68 | 108% |
| **RLIE (Ours)** | **12.23** | **3.40** | **100%** |
| IO Refinement | 8.45 | 2.35 | 69% |

## 5.4. Case Study: Rule Evolution and Representation

To investigate the inner workings of RLIE, we conduct two case studies: one tracing the dynamic trajectory of rule refinement on Retweets, and another analyzing the semantic efficiency of the final rule set on Headlines.

**Dynamics of Rule Evolution.** We track the iterative updates of the rule set in Table 4. The process acts as a continuous error-correction loop. Initial heuristics like urgency

or novelty start with non-zero weights but are effectively pruned to zero as they fail to generalize. Meanwhile, hard examples trigger the generation of more specific semantic features, such as clarity in news contexts, which are then up-weighted. This confirms that RLIE actively performs soft feature selection, evolving from coarse heuristics to robust, task-aligned discriminators.

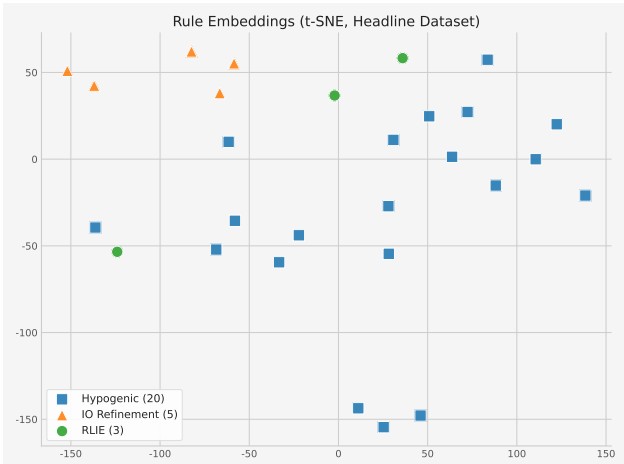

*Figure 3.* t-SNE of rule embeddings. RLIE learns complementary rules while HypoGeniC shows semantic noise and IO Refinement exhibits mode collapse (visualizing all refinement steps).

**Efficiency of Representation.** We visualize the embeddings of the final rule sets for the Headlines task using t-SNE in Figure 3 and inspect the learned rules in Table 5. HypoGeniC exhibits a scattershot distribution, retaining 20 diverse but noisy rules that yield suboptimal accuracy. IO Refinement suffers from mode collapse, where all 5 retained rules cluster tightly and redundantly focus on aggressive imagery. In contrast, RLIE achieves semantic efficiency, converging to just three widely separated rules that capture orthogonal concepts: Surprise and Relatability. Notably,

*Table 4.* **Iterative Rule Refinement Process (Retweet Task).** RLIE refines the rule bank in an error-driven manner. At round $r = 0$, we start from a randomly sampled example; at round $r > 0$, we sample a hard example misclassified by the round $r-1$ model. **[NEW]** newly added rules, **[KEEP]** inherited active rules, and **[DROP]** rules with zeroed weights. Text in **bold** indicates the semantic focus.

**Task:** Predict which of two tweets is retweeted more. **Output:** *First* vs *Second* (0/1).

| Rd | Prompt (Side-by-side) | Rule Bank Update (Full Text) | Metric |
|---|---|---|---|
| 0 | **T1:** ... #House debates issue of #health insurance policies **eliminated** ... 
 **T2:** ... #House voting on bill to **restore** #health insurance policies ... 
 *GT: Second (1)* | **[NEW] 1.** Tweets that use **stronger emotional language or dramatic framing** . . . are more likely to be retweeted. **(w=.035)** 
 **[NEW] 2.** Tweets that include **vivid details, quotes, or specific examples** outperform those that offer general summaries. **(w=.008)** 
 **[DROP] 3.** Tweets that highlight **novelty, urgency, or time-sensitive action** . . . are more likely to be shared. **(w→0)** | Acc: 0.625 
 F1: 0.625 |
| 1 | **T1:** Andrew C. McCarthy: The Scheme behind the Obamacare Fraud ... 
 **T2:** Like all swindles, Obamacare cannot work if its targeted victims ... 
 *GT: Second (1); Prev: First (0)* | **[NEW] 1.** Tweets that **prioritize clarity and immediate comprehension** . . . are favored in news and public-affairs contexts. **(w=.045)** 
 **[KEEP] 2.** Tweets that use **stronger emotional language or dramatic framing** . . . are more likely to be retweeted. **(w=.034)** 
 **[DROP] 3.** Tweets that highlight **novelty, urgency, or time-sensitive action** are more likely to be shared. **(w→0)** | Acc: 0.680 
 F1: 0.679 |
| 2 | **T1:** NEW VID!! :D ***MEET MY DAD!*** ... RT ? :) 
 **T2:** what do you think about the SNEAK PEEK ... ***MEET MY DAD!*** ... 
 *GT: Second (1); Prev: First (0)* | **[NEW] 1.** Tweets that use **personal voice or self-referential framing** . . . are more retweetable when they convey authenticity. **(w=.223)** 
 **[KEEP] 2.** Tweets that **prioritize clarity and immediate comprehension** . . . are favored in news contexts. **(w=.178)** 
 **[DROP] 3.** Tweets with **conversational, informal, or playful tone** are more retweetable than those using formal language. **(w→0)** | Acc: 0.700 
 F1: 0.699 |

*Table 5.* The rule set learned by RLIE. The logistic combiner assigns high importance to Surprise and Relatability features while effectively pruning the Urgency rule ($w = 0$) via $L_1$ regularization.

| Rule ID | Wgt. | Learned Rules | Focus |
|---|---|---|---|
| Rule 2 | 0.039 | *Headlines that frame content as a* **surprising outcome**... *provoke curiosity.* | **Surprise** |
| Rule 0 | 0.005 | *Headlines that frame content as a* **direct, relatable scenario**... | **Relatable** |
| Rule 1 | 0.000 | *Headlines that incorporate a sense of* **urgency**... | *(Pruned)* |

the Urgency rule is correctly identified as ineffective and pruned, mirroring the trajectory seen in the Retweets case. This disentanglement allows RLIE to achieve the highest accuracy with the most compact representation.

## 6. Discussion

We propose a general paradigm for designing inference systems where natural-language rules serve as the basic unit of reasoning. In this setting, future improvements may stem less from injecting global probabilistic details into LLM prompting, and more from shaping a reusable interface that separates LLM-based semantics from probabilistic learning. Compared to traditional predicate-based frameworks, natural-language rules better accommodate open-ended semantics and unstructured data; however, asking an LLM to simultaneously interpret rules and perform global weighted

aggregation can be sensitive to prompt phrasing and instruction competition, potentially hurting stability. A promising direction is to fix the local interface—letting the LLM provide per-rule signals (e.g., judgments or confidence scores) alongside coverage information—and to delegate global consistency, conflict resolution, and uncertainty management to a non-LLM, transparent aggregation module. This module can be further strengthened by interpretable additive models or explicit calibration techniques, enabling robust neuro-symbolic systems that combine the semantic flexibility of LLMs with the rigor of classical statistical models.

## 7. Conclusion

In this work, we introduced RLIE, a unified framework for learning compact, interpretable natural-language rule sets by integrating LLM-based hypothesis generation with a calibrated probabilistic combiner and error-driven refinement. Extensive experiments across diverse real-world tasks and LLM backbones demonstrate that direct inference via the learned logistic combiner yields the most robust and consistent performance, whereas re-injecting rules and weights into LLM prompting often leads to mixed results or degradation. These findings advocate for a principled neuro-symbolic division of labor: LLMs excel at local semantic operations—such as proposing hypotheses and judging applicability—while classical probabilistic models are superior for global selection, weighting, and calibration. This synergy enables the construction of reliable, auditable, and high-performance reasoning systems in natural language.

## Impact Statement

This work advances machine learning by studying how LLMs can generate natural-language decision rules and how classic probabilistic modeling can combine these rules into calibrated, inspectable predictions. A key positive impact is improved transparency and controllability: explicit rule sets and learned weights make it easier to audit model behavior, attribute errors to specific rules, and iteratively refine systems to better match desired policies, which can be valuable for text classification and screening workflows where stakeholders require understandable rationales. The framework may also lower the barrier to building lightweight, modular decision systems by reusing and updating rule sets without retraining large models end-to-end. As with other data-driven approaches, risks include learning rules that encode biases or spurious correlations present in data, or over-trusting rule-based rationales in high-stakes settings. These risks can be mitigated through careful dataset curation and documentation, evaluation across subpopulations and robustness tests, and maintaining human oversight when deploying or updating rule sets.

## Acknowledgments

This work was supported by the Guangdong Basic and Applied Basic Research Foundation (Grant No. 2026A1515011579), the HKUST-HKUST(GZ) 1+1+1 Joint Funding Program (Grant No. C_2025_031), and the Guangzhou-HKUST(GZ) Joint Funding Program (Grant No. 2023A03J0008), Education Bureau of Guangzhou Municipality. This work was also supported by Jiangsu Industrial Technology Research Institute (JITRI) and Wuxi National High-Tech District (WND).

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

# A. More Details

In this section, we provide additional information regarding the baseline methods used for comparison and the specific validation procedures employed in our experiments.

## A.1. Baselines

To contextualize the performance of **RLIE**, we compare it against two state-of-the-art LLM-based rule learning frameworks:

- **IO Refinement**: Proposed by Qiu et al. (2023), this method follows a "propose-select-refine" loop. It begins by generating candidate rules and iteratively refines a single best hypothesis based on feedback from its performance on a set of examples.

- **HypoGeniC**: Introduced by Zhou et al. (2024), this framework maintains a dynamic library of hypotheses. It uses a reward mechanism inspired by multi-armed bandits to balance the exploration of new rules and the exploitation of well-performing ones, updating its rule set based on performance on training instances.

## A.2. Validation

Our experimental setup follows a standardized procedure, but certain details regarding the use of the validation set for baselines and the handling of smaller datasets warrant clarification.

For datasets where the total number of available samples was less than the specified 200 for training and 200 for validation, we utilized all available data for the respective splits.

The baseline methods handle the validation set differently according to their original designs. Since the algorithms for **IO Refinement** and **Zero-shot Generation** do not inherently use a validation set during their rule generation phase, we implemented a final selection step for a fair comparison: after these methods completed their final iteration, we selected the single rule that achieved the highest performance on our designated validation set to be used for testing. In contrast, the **HypoGeniC** algorithm does not involve a validation set in its update loop; its mechanism for updating and pruning the hypothesis library relies on rewards calculated from performance on the training batches. Therefore, for HypoGeniC, the validation set was not used during its training and refinement process.

## A.3. Dataset

**Deception Detection (Reviews)**. The objective is to distinguish between genuine and deceptive hotel reviews. Deceptive reviews, often written by paid individuals, may appear plausible but typically lack specific details. The primary challenge lies in identifying subtle linguistic cues, such as exaggerated sentiment or vague descriptions, which are difficult for even human evaluators to discern reliably.

**Persuasive Argument Prediction (Dreaddit)**. The objective is to detect mental stress signals from Reddit posts across different communities. These posts, sourced from social media, often contain complex personal narratives that reflect an individual's psychological state. The primary challenge lies in investigating specific linguistic features indicative of mental stress, identifying subtle patterns in the text that effectively signal the presence of distress.

**News Headline Engagement (Headlines)**. Given two headlines for the same news event, the model must predict which one is more likely to attract clicks. The task examines how linguistic choices in news writing—such as novelty, specificity, or emotional framing—influence reader behavior. The often minimal difference between headlines makes this task highly challenging.

**Paper Citations (Citations)**. This task involves predicting whether an academic paper will achieve a high citation count based on its title and abstract. It focuses on features related to academic impact, such as the generality of the research problem, the novelty of the contributions, or the timeliness of the research topic. Unlike social media tasks, this task emphasizes long-term value signals in academic writing.

**AI-generated Content Detection (LLM Detect)**. This task requires determining whether a given text was written by a human or generated by an LLM. The dataset contains human- and machine-written texts based on the same prompts, evaluating a model's ability to capture stylistic, structural, or semantic differences indicative of AI-generated content and formulate them as interpretable rules.

**Retweets (Retweets).** The input is a pair of tweets, and the task is to predict which one will be retweeted more. Influential factors include a tweet's conciseness, emotional intensity, and the mention of specific individuals or organizations. Due to the highly stochastic nature of social media propagation, this task places stringent demands on the explanatory power and stability of the learned rules.

### A.4. Hyperparameter Settings

**Iterative Refinement.** To balance performance and computational cost, we set the maximum number of refinement iterations $R_{max} = 20$. The early stopping mechanism uses a patience of $p = 3$ rounds with a minimum improvement threshold $\delta = 1 \times 10^{-3}$ on the validation set accuracy.

**Logistic Regression.** We implement the logistic combiner using the `scikit-learn` library. Hyperparameters for Elastic Net are selected via 5-fold stratified cross-validation on $\mathcal{S}_{train}$. We perform a grid search for the inverse regularization strength $C$ (30 logarithmically spaced values in $[10^{-3}, 10^1]$) and the L1 ratio $\alpha$ (30 linearly spaced values in $[0.01, 0.99]$).

**LLM Generation.** For the Rule Generation and Refinement stages, we prioritize stability and reproducibility. We set the generation temperature to $1 \times 10^{-5}$. The sampling process uses a batch size of 20 training samples, generating 5 rules per API call, with a maximum token limit of 8000.

## B. Parameter Study

*Table 6.* Sensitivity analysis of the coverage threshold $\gamma$ on the Headline dataset (Backbone: DeepSeek-V3). Performance remains robust for $\gamma \in [0.1, 0.5]$.

| Coverage Threshold ($\gamma$) | Accuracy | F1 Score |
|:---:|:---:|:---:|
| 0.1 | 66.7 | 66.7 |
| 0.2 | 67.0 | 67.0 |
| 0.3 | **67.3** | **67.2** |
| 0.4 | 66.5 | 66.5 |
| 0.5 | 66.5 | 66.5 |
| 0.6 | 65.5 | 65.5 |
| 0.7 | 66.5 | 66.5 |
| 0.8 | 65.6 | 65.6 |
| 0.9 | 64.3 | 64.2 |

## C. Calibration Analysis

To validate our claim that the logistic combiner provides calibrated probabilistic outputs, we evaluate the **Expected Calibration Error (ECE)** (Guo et al., 2017) of RLIE compared to the rule-injection prompting baseline (E2).

### C.1. Methodology

We compute ECE using 10 equal-width bins. For both methods, ECE measures the weighted average difference between the predicted confidence and the empirical accuracy: $\text{ECE} = \sum_{m=1}^{M} \frac{|B_m|}{N} |\text{acc}(B_m) - \text{conf}(B_m)|$.

- **RLIE (Ours):** The probability is directly derived from the logistic regression combiner: $p(y = 1|x) = \sigma(\beta_0 + \sum w_j r_j(x))$. Since logistic regression minimizes the negative log-likelihood, it naturally aligns predicted probabilities with empirical frequencies.

- **Prompting (E2):** Since standard prompting yields binary text, we adapt it to produce probabilistic outputs for fair comparison. We constrain the LLM (DeepSeek-V3.2) to output a single token (e.g., 'A' or 'B') and extract the **token log-probabilities**. The probability of the positive class is computed via softmax: $p = \frac{\exp(\ell_{pos})}{\exp(\ell_{pos}) + \exp(\ell_{neg})}$.

## C.2. Results

Table 7 reports the mean ECE and standard deviation over 3 runs. **RLIE achieves consistently lower ECE scores (better calibration) on 5 out of 6 datasets.** Notably, on tasks like *Retweets* and *Reviews*, Prompting exhibits high calibration error ($> 0.45$), indicating severe overconfidence (predictions are often close to 1.0 even when wrong). In contrast, RLIE maintains an ECE around 0.11, demonstrating that the learned weights serve as reliable uncertainty estimates.

*Table 7.* Comparison of Expected Calibration Error (ECE) on the test set (Lower is better). RLIE significantly outperforms Prompting (E2) on most tasks, providing more reliable confidence estimates.

| Dataset | RLIE (Ours) ECE $\downarrow$ | Prompting (E2) ECE $\downarrow$ |
|---|---|---|
| Retweets | $\mathbf{0.116} \pm 0.041$ | $0.480 \pm 0.040$ |
| Reviews | $\mathbf{0.113} \pm 0.022$ | $0.511 \pm 0.065$ |
| Headlines | $\mathbf{0.104} \pm 0.029$ | $0.371 \pm 0.034$ |
| Dreaddit | $\mathbf{0.226} \pm 0.054$ | $0.368 \pm 0.050$ |
| Citations | $\mathbf{0.179} \pm 0.048$ | $0.494 \pm 0.261$ |
| LLM Detect | $0.252 \pm 0.052$ | $\mathbf{0.187} \pm 0.051$ |

# D. LLM Usage

Large Language Models (LLMs) were used as an assistive tool in preparing this manuscript. The core intellectual contributions, including the research ideas, experimental design, and analysis, are entirely the work of the human authors. The LLM's role was limited to language polishing and coding assistance. The authors have reviewed and edited all LLM-generated content and take full responsibility for the final manuscript.

# E. Prompts

Here we are providing the prompts we are using to conduct all the experiments. For the clarity of illustration, here we are only showing prompts for the Retweet task, while all the prompts can be access after this paper is accepted.

```
multi_content: |
    The first tweet: ${first_tweet}
    The second tweet: ${second_tweet}
    Final answer: The ${label} tweet got more retweets.
```

*Figure 4.* Prompt for providing observations.

```
system: |-
    You are a social media expert. You are an expert at determining which tweet will
        be retweeted more.
    Given a set of observations, you want to generation hypotheses that will help
        predict which tweet out of a pair of tweets is more likely to be retweeted.
    Please note that the paired tweets are about the same content and are posted by
        the same user, so you should focus on the wording difference between the two
        tweets in each pair.
    Please propose ${num_hypotheses} possible hypotheses.
    Please generate them in the format of:
    1. [hypothesis]
    2. [hypothesis]
    ...
    ${num_hypotheses}. [hypothesis].
    Please make the hypotheses general enough to be applicable to new observations.
user: |-
    We made some observations:
    ${observations}
    Generate hypotheses that are useful for predicting which tweet out of a pair of
        tweets is more likely to be retweeted.
    Please note that the paired tweets are about the same content and are posted by
        the same user, so you should focus on the wording difference between the two
        tweets in each pair.
    Please propose ${num_hypotheses} possible hypotheses.
    Please generate them in the format of:
    1. [hypothesis]
    2. [hypothesis]
    ...
    ${num_hypotheses}. [hypothesis].
    Proposed hypotheses:
```

*Figure 5.* Prompt for the first iteration of rule generation.

```
system: |-
  You are a social media expert focused on maximizing retweet engagement.
  Given misclassified tweet pairs and prior hypotheses, your goal is to rethink and
      propose ${num_hypotheses} new, more accurate hypotheses about which tweet in a
      pair will earn more retweets.
  Please note that the paired tweets share the same content and author, so concentrate
       on wording differences, framing, and presentation.
  Generate the hypotheses in the format of:
  1. [hypothesis]
  2. [hypothesis]
  ...
  ${num_hypotheses}. [hypothesis].
  Please make the hypotheses general enough to be applicable to new observations.
user: |-
  We have tweet pairs that previous hypotheses predicted incorrectly:
  ${observations}

  Here are some of the prior hypotheses for reference:
  ${hypotheses_text}

  Please generate new hypotheses that better capture which tweet in each pair will get
       more retweets.
  You may refine the previous hypotheses (like tightening conditions, adding
      exceptions, or rephrasing), or introduce new hypotheses to cover new, distinct
      angles when prior ones are insufficient or misaligned.

  Propose ${num_hypotheses} possible hypotheses.
  Generate them in the format of 1. [hypothesis], 2. [hypothesis], ... ${
      num_hypotheses}. [hypothesis].
  Proposed hypotheses:
```

*Figure 6.* Prompt for iterative refinement.

```
system: |-
  You are a social media expert.
  Given a pair of tweets, you are asked to predict which tweet will be retweeted more.
  Please note that the paired tweets are about the same content and are posted by the
      same user, so you should focus on the wording difference between the two tweets.
  From past experiences, you learned a pattern.
  Now, at each time, you should apply a learned pattern to a pair of tweets and
      determine which one will get more retweets.
  Give an answer. Respond with exactly one of: first, second, or not applicable.
  Give your final answer in the format of {Final answer: first} or {Final answer:
      second}.
user: |-
  Pattern: ${hypothesis}
  The first tweet: ${first_tweet}
  The second tweet: ${second_tweet}

  Given the pattern you learned above, predict which one of the two tweets will get
      more retweets.
  Think step by step.
  First step: Consider if the pattern can be applied to analyze the textual difference
      between the two tweets.
  Third step: Based on the pattern, which tweet is more likely to get more retweets?
      If it does not apply, say so explicitly.
  Final step: Give your final answer in the format of {Final answer: first}, {Final
      answer: second} or {Final answer: not applicable}.
  Final answer:
```

*Figure 7.* Prompt for single hypothesis inference.

```
system: |-
  You are a social media expert.
  Given a pair of tweets, you are asked to determine which will get more retweets.
  From past experiences, you learned some patterns.
  You need to determine whether each of the patterns holds for the current pair of
      tweets, and also predict which tweet will get more retweets.
  Give your final answer in the format of {Final answer: first} or {Final answer:
      second}.
user: |-
  Our learned patterns: ${hypotheses}
  The first tweet: ${first_tweet}
  The second tweet: ${second_tweet}

  Given the patterns you learned above, predict which one will get more retweets.
  Think step by step.
  First step: Think about which patterns can be applied to these tweets.
  Second step: Based on the applicable patterns, which tweet is likely to get more
      retweets?
  Give your final answer in the format of {Final answer: first} or {Final answer:
      second}.
```

*Figure 8.* Prompt for "LLM + Rules" style inference.

```
system: |-
  You are a social media expert.
  Given a pair of tweets, you are asked to determine which will get more retweets.
  We trained a linear regression model on the training set to obtain a collection of
      weighted patterns plus a bias term. The learned weight reflects how strongly the
      pattern contributes to predicting "first" vs "second".
  Review the weighted hypotheses, consider how the bias interacts with them, and use
      the regression model's suggested label only as a reference.
  Give your final answer in the format of {Final answer: first} or {Final answer:
      second}.
user: |-
  Our learned weighted patterns (the weight's magnitude reflects the pattern's
      importance):
  ${weighted_hypotheses}

  Bias term (a constant offset added regardless of pattern activations; a positive
      bias means the model is overall more inclined toward ${pos_label}, while a
      negative bias leans toward ${neg_label}):
  ${bias}

  The first tweet: ${first_tweet}
  The second tweet: ${second_tweet}

  Given the patterns you learned above, predict which one will get more retweets.
  Think step by step.
  First step: Think about which patterns can be applied to these tweets.
  Second step: Decide which tweet is likely to get more retweets, you can used the
      weighted patterns and bias as reference.
  Final step: give your final answer in the format of {Final answer: first} or {Final
      answer: second}.
```

*Figure 9.* Prompt for "LLM + Rules + Weights" style inference.

```
system: |-
  You are a social media expert.
  Given a pair of tweets, you are asked to determine which will get more retweets.
  We trained a linear regression model on the training set to obtain a collection of
      weighted patterns plus a bias term. The learned weight reflects how strongly the
      pattern contributes to predicting "first" vs "second". The regression model also
      outputs a referenced label for the review, which should be treated as a
      suggestion.
  Review the weighted hypotheses, consider how the bias interacts with them, and use
      the regression model's suggested label only as a reference.
  Give your final answer in the format of {Final answer: first} or {Final answer:
      second}.
user: |-
  Our learned weighted patterns (the weight's magnitude reflects the pattern's
      importance):
  ${weighted_hypotheses}

  Bias term (a constant offset added regardless of pattern activations; a positive
      bias means the model is overall more inclined toward ${pos_label}, while a
      negative bias leans toward ${neg_label}):
  ${bias}

  The first tweet: ${first_tweet}
  The second tweet: ${second_tweet}

  We have used the regression model to get a referenced label: ${model_prediction}

  Given the patterns you learned above, predict which one will get more retweets.
  Think step by step.
  First step: Think about which patterns can be applied to these tweets.
  Second step: Decide which tweet is likely to get more retweets, you can used the
      weighted patterns, bias, predicted label as reference.
  Final step: give your final answer in the format of {Final answer: first} or {Final
      answer: second}.
```

*Figure 10.* Prompt for "LLM + Rules + Weights + Linear Prediction" style inference.

