# OpenReview forum: "RLIE: Rule Generation with Logistic Regression, Iterative Refinement, and Evaluation for Large Language Models"
_ICML.cc/2026/Conference — ICML 2026 regular_

### Official Review · Reviewer_RM1Z · 2026-02-25

**Soundness:** 3
**Presentation:** 3
**Significance:** 3
**Originality:** 2
**Overall Recommendation:** 4
**Confidence:** 3

**Summary:**

This paper proposes RLIE, a framework for learning interpretable natural-language rule sets by combining large language models with a calibrated probabilistic aggregation module. The goal is to construct compact, reusable rule-based systems for binary classification tasks over natural-language inputs while maintaining both interpretability and predictive reliability.

The framework consists of four stages. First, an LLM generates candidate natural-language rules from a subset of the training data. Each rule is then applied to all training instances through an LLM-based judge that determines whether the rule supports the positive class, negative class, or abstains. Second, these rule-application signals are used as features in a regularized logistic regression model, which learns sparse and calibrated rule weights to perform global aggregation. Third, the method iteratively refines the rule set by identifying hard examples under the current model and prompting the LLM to revise or extend the rules accordingly, followed by rule pruning based on validation performance. Finally, the learned rule set and weights are evaluated under multiple inference protocols, comparing direct probabilistic aggregation with rule-injection prompting strategies.

Experiments are conducted on six real-world text classification tasks using multiple LLM backbones. The results show that the explicit logistic combiner provides more stable and better-calibrated performance than directly prompting the LLM to aggregate rules, even when rule weights and reference predictions are injected into the prompt. The authors argue that these findings support a principled division of labor: LLMs are effective for semantic rule generation and local rule application, while classical probabilistic models are better suited for global aggregation and calibration.

**Compliance With Llm Reviewing Policy:**

Affirmed.

**Final Justification:**

The author's second round of rebuttal basically answered my question. I have decided to raise the rating to a more positive one.

**Key Questions For Authors:**

**Q1. Robustness of the LLM-based rule judge.**

In Section 3.1, rule activations are entirely determined by an LLM judge that outputs ternary signals ($-1, 0, +1$). Since the logistic regression model is trained solely on these signals, the stability of this judge is central to the framework. Could the authors provide empirical evidence on the robustness of rule activations across different LLM backbones or decoding configurations (e.g., cross-backbone agreement rates or activation variance)?

If the rule-application stage is shown to be stable and consistent across backbones, this would substantially strengthen the technical soundness of the framework. Conversely, if rule activations are highly sensitive, it would suggest that the learned rule weights may depend strongly on the specific LLM configuration, limiting reproducibility and generality.

**Q2. Validation usage and fairness of comparisons.**

Appendix A.2 indicates that RLIE uses the validation set for hyperparameter tuning, rule pruning, and early stopping, while certain baselines (e.g., HypoGeniC) do not involve validation in their update loop. Could the authors clarify whether any additional experiments were conducted under matched validation usage protocols, or provide ablations quantifying how much performance gain comes from validation-based selection?

A clearer analysis here could meaningfully affect my evaluation of the empirical soundness. If performance remains competitive under strictly matched model-selection conditions, confidence in the reported gains would increase. If validation usage accounts for a significant portion of the improvement, the comparative advantage would need to be interpreted more cautiously.

**Q3. Scalability and rule capacity.**

The rule capacity is fixed at $H=10$ (Section 4.3), and each refinement iteration adds a small number of rules. Have the authors evaluated performance and computational behavior under larger rule banks? In particular, how does prediction quality and token cost scale as $H$ increases?

Given that rule application requires LLM calls for each rule-instance pair, scalability may become a bottleneck in more complex settings. Evidence that the method remains stable and computationally manageable for larger rule sets would increase confidence in its broader applicability beyond small-capacity systems.

**Limitations:**

Yes

**Strengths And Weaknesses:**

**Soundness**

The paper presents a technically coherent and well-specified framework. The four-stage pipeline is clearly described, and the use of Elastic Net regularization with validation-based model selection is appropriate. The empirical evaluation spans six real-world datasets and multiple LLM backbones, and the comparison of inference protocols (E1--E4) directly supports the central empirical claim that explicit probabilistic aggregation is more stable than prompt-based rule injection.

However, several aspects limit the depth of technical analysis. First, rule application relies entirely on an LLM-based judge that outputs ternary signals (Section3.1), yet no analysis is provided on the stability or variability of these judgments across backbones or decoding conditions. Since the logistic combiner operates solely on these signals, the absence of a robustness study leaves a key component unexamined. Second, the iterative refinement mechanism is heuristic and repeatedly leverages validation performance for pruning and early stopping, but the paper does not discuss convergence behavior, potential validation overfitting, or stability of rule evolution. Third, validation usage is asymmetric: RLIE employs the validation set for hyperparameter tuning, rule pruning, and early stopping, whereas certain baselines do not use validation during their update loop. While consistent with original implementations, this difference is not explicitly analyzed in terms of fairness. Finally, scalability is only partially addressed. Although token usage is reported, the rule capacity is fixed at $H=10$, and there is no investigation of performance or computational behavior under larger rule sets.

**Presentation**

The paper is clearly written and well structured. The conceptual framing around a ``division of labor'' between LLM-based semantic induction and probabilistic aggregation provides a coherent narrative. The inference protocols are precisely defined and experimentally compared, and implementation details and prompts are documented in the appendix, supporting transparency.

That said, certain design choices are not fully justified. The selection of logistic regression as the aggregation mechanism is motivated primarily by calibration arguments, but no comparison is provided against alternative additive or interpretable aggregation models. Moreover, while related work is discussed, the distinction from prior multi-hypothesis refinement systems remains largely empirical rather than formally articulated, which somewhat weakens the clarity of methodological differentiation.

**Significance**

The paper addresses a practically relevant problem: how to construct interpretable and calibrated rule-based systems from LLM-generated natural-language hypotheses. The systematic evaluation of inference strategies demonstrates that injecting rule weights into prompts does not consistently improve performance, and the calibration analysis provides useful empirical insight into reliability differences. These findings may influence future work on hybrid neuro-symbolic systems.

Nevertheless, the empirical scope is limited to binary text classification tasks, and no experiments are conducted on multi-class, structured, or compositional reasoning problems. As a result, the demonstrated impact remains primarily within interpretable classification settings rather than broader reasoning domains. In addition, performance gains over strong baselines are moderate on several datasets, suggesting that the practical advantage may be task-dependent.

**Originality**

The work offers a clear architectural perspective by explicitly separating semantic rule induction from global probabilistic aggregation and empirically validating this separation through multiple inference protocols. The systematic comparison of aggregation strategies provides insight into limitations of prompt-based weighted reasoning and contributes to understanding current LLM behavior.

However, the individual components of the framework—LLM-based rule generation, hard-example mining, Elastic Net logistic regression, and rule pruning—are established techniques. The contribution lies primarily in their integration and empirical evaluation rather than in introducing new algorithms, objectives, or theoretical advances. Thus, the originality is mainly at the system-design and empirical-insight level rather than at the methodological level.

---

> ### Author Rebuttal · Authors · 2026-03-30
>
> We thank the reviewer for recognizing RLIE's **technical coherence and clarity**; we reply below.
>
> > **Weakness 1:** *The originality seems to lie mainly at the system-design level.*
>
> The real contribution is at the **framework level**: a predictive rule-learning loop that couples judging, probabilistic aggregation, pruning, and refinement, and we also provide a detailed evaluation of LLMs’ ability to understand probabilistic rules.  For more details, please see **Reviewer zhQp, Weakness 1**.
>
> > **Weakness 2:** *The stability of the rule judge is not analyzed.*
>
> We added a **judge-robustness** analysis. For more details, please see **Question 1** below.
>
> > **Weakness 3:** *Iterative refinement is heuristic and lacks stability analysis.*
>
> We added a **refinement-stability** analysis.
>
> **(1) Iterative refinement is effective.** On all 6 tasks, the **best mean validation accuracy** is higher than **Round 0**, with an average improvement of about **0.0576**. For more details, please see **Reviewer eooD, Question 3**.
>
> **(2) Later stages are usually stable.** Local validation fluctuations around the selected round are only **0.0067–0.0317**, which looks more like a local plateau than unconstrained drift.
>
> **(3) Rule evolution itself also stabilizes.** On **headline_binary**, the average weight change decreases from **0.1282** to **0.0515** (**Kendall distance 0.1048**). The full Round-0 vs. best-round comparison is provided in [Table 4](https://anonymous.4open.science/r/supp-tables-5C58/RLIE_rebuttal_external_tables.md). A brief summary is also given in **Reviewer oqmB, Weakness 2**.
>
> > **Weakness 4:** *The use of validation needs a fairer comparison.*
>
> We disabled **validation-based best-round selection** and directly used **the final round** instead. Performance remains **largely stable**, with only mild drops, while RLIE still stays ahead of all baselines, making the comparison fairer. For more details, please see **Question 2** below.
>
> > **Weakness 5:** *Scalability under larger rule banks is not analyzed.*
>
> The default fixed capacity in the main paper was used for **controlled comparison**.
>
> Our added analysis shows that **very small rule banks are insufficient to fully realize RLIE’s advantage**, while performance becomes relatively stable once the rule bank reaches a larger size. Cost also does not increase linearly with the number of rules; in some settings, it may even decrease because of earlier stopping.
>
> For more details, see **Question 3** below and **Reviewer eooD's Question 3**.
>
> > **Weakness 6:** *LR is not compared with other interpretable aggregators.*
>
> We additionally compare against **unweighted majority voting** and **Zhong et al.**, and both comparisons show robust advantages.
>
> For more details, please see **Reviewer oqmB, Question 1**, and **Reviewer zhQp's Weakness 2**.
>
> > **Weakness 7:** *The empirical scope is limited to binary classification.*
>
> The iterative hypothesis-generation framework of RLIE itself is general, and in future work we will extend it to **multi-class classification, regression, and multi-step reasoning** tasks.
>
> > **Question 1:** *How robust are rule activations across different backbones?*
>
> To study robustness, we fix the **rule set** and **logistic readout** learned by **DeepSeek V3.2**, and only replace the **judge backbone** responsible for rule activation.
>
> Across all 6 tasks, when using **DeepSeek-V3.2-Exp**, **qwen3-235b-a22b-instruct-2507**, and **qwen3-next-80b-a3b-instruct** as judges, pairwise **non-abstain agreement** ranges from **0.7964** to **0.9204**, and **prediction agreement** ranges from **0.7267** to **0.8689**. More complete pairwise results are reported in [Table 7](https://anonymous.4open.science/r/supp-tables-5C58/RLIE_rebuttal_external_tables.md).
>
> > **Question 2:** *How important is validation-based selection?*
>
> We disabled **validation-based round selection**; in practice, this also removes **early stopping** and lets refinement run to the fixed maximum number of rounds. Over the 6 tasks, the average **best-vs-last** gap is about **0.0168** (**validation ACC**) and **0.0178** (**validation F1**), while the overall performance still remains clearly ahead of the main baselines. Full per-task gaps are reported in [Table 6](https://anonymous.4open.science/r/supp-tables-5C58/RLIE_rebuttal_external_tables.md).
>
> > **Question 3:** *How do quality and token cost change when the rule-bank size increases?*
>
> On the **Headlines** dataset, we specifically ran a **rule capacity ablation** with **2 / 3 / 5 / 10 / 20 / 30** rules, and jointly reported both **performance** and **total token usage**. Very small rule capacity limit RLIE’s advantage; performance stabilizes at larger capacity; and token cost is not strictly monotonic, because some small-capacity settings may actually trigger longer refinement. For full numbers, see **Reviewer eooD's Question 3** and [Table 2](https://anonymous.4open.science/r/supp-tables-5C58/RLIE_rebuttal_external_tables.md).

---

> > ### Author Rebuttal · Reviewer_RM1Z · 2026-04-03
> >
> > I appreciate the authors’ detailed rebuttal, which effectively addresses my concerns about rule judge robustness, iterative refinement stability, validation fairness, and rule bank ablation. These responses mitigate my soundness-related worries. I am therefore maintaining my original score of 3, as the system-level novelty and evaluation scope remain unchanged. To provide an opportunity for further clarification, I would encourage the authors to comment on whether the rule bank scalability trends observed on the Headlines dataset (2–30 rules) generalize to the other tasks, and whether extreme capacity limits affect convergence or token cost. Addressing this could strengthen confidence in RLIE’s practical scalability.

---

> > > ### Author Response · Authors · 2026-04-07
> > >
> > > We thank the reviewer for the further careful assessment, and we also appreciate your recognition that our previous additions on **judge robustness, iterative refinement stability, validation fairness**, and the initial **rule-bank ablation** have already helped address the soundness-related concerns. To further respond to your follow-up question on **practical scalability**, beyond the previously reported `2–30`-rule results on **Headlines**, we extended the **rule-bank size analysis to the other five datasets** and systematically summarized the resulting **ACC / F1 / total tokens** across capacities.
> > >
> > > Overall, the added results are broadly consistent with the trend observed on **Headlines**, and further suggest that this scalability pattern is not specific to a single dataset: **very small rule banks often fail to fully realize RLIE’s advantage, while performance generally improves as the number of rules increases and gradually saturates at larger capacities.** Averaged over all six tasks, Avg ACC / F1 improves from `0.686 / 0.683` at `H=2` to `0.755 / 0.752` at `H=20`, and then remains essentially flat at `H=30` (`0.754 / 0.752`). At the same time, **token cost generally increases with capacity** (from `8.02M` to `24.58M` on average), showing that larger rule banks are not free; however, this relationship is **not strictly monotonic** on individual tasks, since some moderate-capacity settings even use fewer tokens than smaller ones. This suggests that capacity affects not only per-round rule-application cost, but also the overall **refinement dynamics**. We summarize the added results below.
> > >
> > > | # Rules | Avg ACC | Avg F1 | Avg Total Tokens (M) |
> > > |---|---:|---:|---:|
> > > | 2  | 0.686 | 0.683 | 8.02 |
> > > | 3  | 0.718 | 0.716 | 7.57 |
> > > | 5  | 0.724 | 0.722 | 13.66 |
> > > | 10 | 0.735 | 0.732 | 12.23 |
> > > | 20 | 0.755 | 0.752 | 21.32 |
> > > | 30 | 0.754 | 0.752 | 24.58 |
> > >
> > > Below we provide the detailed results for the added five tasks (the **Headlines** results were already reported in the previous round and are not repeated here):
> > >
> > > ### **Reviews**
> > >
> > > | # Rules | 2 | 3 | 5 | 10 | 20 | 30 |
> > > |---|---:|---:|---:|---:|---:|---:|
> > > | Test ACC | 0.696 | 0.708 | 0.712 | 0.709 | 0.736 | 0.734 |
> > > | Test F1 | 0.691 | 0.708 | 0.711 | 0.707 | 0.735 | 0.733 |
> > > | Total Tokens (M) | 18.62 | 10.93 | 20.73 | 20.04 | 41.90 | 48.20 |
> > >
> > > ### **Dreaddit**
> > >
> > > | # Rules | 2 | 3 | 5 | 10 | 20 | 30 |
> > > |---|---:|---:|---:|---:|---:|---:|
> > > | Test ACC | 0.800 | 0.810 | 0.810 | 0.823 | 0.824 | 0.824 |
> > > | Test F1 | 0.800 | 0.810 | 0.810 | 0.823 | 0.824 | 0.824 |
> > > | Total Tokens (M) | 5.56 | 10.93 | 5.71 | 7.10 | 18.18 | 20.30 |
> > >
> > > ### **Citations**
> > >
> > > | # Rules | 2 | 3 | 5 | 10 | 20 | 30 |
> > > |---|---:|---:|---:|---:|---:|---:|
> > > | Test ACC | 0.598 | 0.615 | 0.631 | 0.646 | 0.719 | 0.713 |
> > > | Test F1 | 0.586 | 0.604 | 0.621 | 0.630 | 0.716 | 0.710 |
> > > | Total Tokens (M) | 2.25 | 2.55 | 2.96 | 3.42 | 6.60 | 7.40 |
> > >
> > > ### **LLM Detect**
> > >
> > > | # Rules | 2 | 3 | 5 | 10 | 20 | 30 |
> > > |---|---:|---:|---:|---:|---:|---:|
> > > | Test ACC | 0.833 | 0.870 | 0.860 | 0.907 | 0.912 | 0.913 |
> > > | Test F1 | 0.833 | 0.870 | 0.860 | 0.907 | 0.912 | 0.913 |
> > > | Total Tokens (M) | 10.73 | 10.63 | 34.52 | 22.86 | 29.60 | 33.20 |
> > >
> > > ### **Retweets**
> > >
> > > | # Rules | 2 | 3 | 5 | 10 | 20 | 30 |
> > > |---|---:|---:|---:|---:|---:|---:|
> > > | Test ACC | 0.556 | 0.662 | 0.646 | 0.657 | 0.669 | 0.670 |
> > > | Test F1 | 0.552 | 0.662 | 0.646 | 0.656 | 0.668 | 0.669 |
> > > | Total Tokens (M) | 3.66 | 4.64 | 6.66 | 9.36 | 13.70 | 15.90 |
> > >
> > > These added results provide more direct empirical evidence for RLIE’s **practical scalability**, and further indicate that the trend previously observed on **Headlines** transfers reasonably consistently to the other tasks. We hope these additions more fully address your remaining questions about cross-task generalization and the impact of extreme capacities on performance, token cost, and refinement dynamics; if these new analyses have helped alleviate your remaining concerns, we would be very grateful if you would reconsider your evaluation of the paper.

---

### Official Review · Reviewer_oqmB · 2026-03-08

**Soundness:** 4
**Presentation:** 4
**Significance:** 4
**Originality:** 3
**Overall Recommendation:** 5
**Confidence:** 4

**Summary:**

The paper proposes a text classification method in which LLMs propose features for logistic regression (specifically elasticNet L1+L2 regression) and show the new method outperforms plausible baselines on 6 benchmarks with three backbone LLMs from two LLM families (DeepSeek and Qwen).  The feature generation is iterative, based on sampling examples that are hard-to-classify with the most recent classifier.

**Compliance With Llm Reviewing Policy:**

Affirmed.

**Final Justification:**

Based on the rebuttal I have raised my score to accept

**Key Questions For Authors:**

Can you clarify and/or quantify the contribution beyond that made by Zhong et al?

**Limitations:**

yes

**Strengths And Weaknesses:**

Strengths:
 - Well written and motivated
 - Solid experimental results, with strong gains on 2/6 benchmarks and a consistent improvement over the baselines.

Weaknesses:
 - A very similar system was proposed in "Explaining Datasets in Words", Zhong et al NeurIPS 2024.  Zhong et al describe a general scheme to extend many learning models with natural language predicates, but one of the concrete cases they consider is logistic regression (without regularization), and their approach also iteratively refines LLM-produced predicates by resampling, and it clearly addresses the core question here of "probabilistic composition of natural-language rules"
 - Some parts of the system aren't experimentally supported by ablation studies: e.g., how important is the L1/L2 regularization? the iterative rule refinement? the rule pruning (given than L1 regression is used, wouldn't that tend to zero out unimportant rules?) the limits on rule capacity?

Zhong et al paper is https://proceedings.neurips.cc/paper_files/paper/2024/hash/90c4537a301e9545bb4c60219f2992b1-Abstract-Conference.html - if this prior work didn't exist I think the paper would be a solid accept, but given the closely related prior work I feel some sort of experimental comparison is needed.

---

> ### Author Rebuttal · Authors · 2026-03-30
>
> We sincerely thank the reviewer for the careful evaluation and for recognizing that our paper is **well written**, **well motivated**, and supported by **solid empirical results**. Below, we address your concerns one by one.
>
> > **Weakness 1:** *The comparison to Zhong et al. (NeurIPS 2024) is missing.*
>
> Thank you for pointing out this related work. We agree that Zhong et al. and RLIE share some high-level similarity, but they differ substantially in both **problem formulation** and **concrete mechanism**. We will clarify this relationship more carefully in the revision. For more details, please see **Question 1** below.
>
> > **Weakness 2:** *Several components still lack direct ablations, including regularization, pruning, refinement, and capacity control.*
>
> Thank you for this suggestion. Our relevant analysis is as follows.
>
> **On regularization and pruning.**
> For **L1/L2 regularization** and **rule pruning**, these are not independent small tricks that can be cleanly switched on or off one by one. Instead, together they serve as a coupled mechanism for stabilizing refinement and controlling effective capacity. Removing them would cause uncontrolled growth of the rule bank and make the experiment fail. We will explain this role more clearly in the revision.
>
> **On refinement and rule-bank capacity H.**
> Across all 6 tasks, the best refined results improve F1 by **0.055 on average** over the initial version, which shows the value of iteration. Meanwhile, the results remain fairly stable as **H varies from 2 to 30**. For more details, please see **Reviewer RM1Z, Weakness 2**, and **Reviewer eooD, Question 3**.
>
> > **Question 1:** *Can you clarify and quantify your contribution relative to Zhong et al.?*
>
> Beyond the broad template of *“natural-language units + learnable readout,”* RLIE differs from Zhong et al. in four concrete respects.
>
> **Differences at the formulation level.**
>
> **(1) RLIE learns predictive rules rather than predicates.**
> Zhong et al. learn natural-language predicates, usually corresponding to a **fixed predicate bank**, followed by a linear readout. RLIE instead learns **directional predictive rules**, and the rule set itself evolves **adaptively** rather than being fixed in advance. This yields stronger expressivity and is more directly aligned with the prediction objective.
>
> **(2) RLIE’s rules can abstain.**
> Our rules are not forced to output a binary judgment for every sample. When a condition has no substantive applicability, a rule can return **not applicable / abstain**. This is important because natural-language rules often have selective applicability rather than universal applicability.
>
> **Differences at the optimization level.**
>
> **(1) RLIE jointly optimizes the whole rule set rather than replacing one predicate slot at a time.**
> Zhong et al. maintain a fixed-size predicate bank and replace the weakest predicate in each refinement round. By contrast, RLIE first proposes new candidate rules, then **re-optimizes the entire rule set** under sparsity and capacity constraints.
>
> **(2) RLIE uses probability-guided closed-loop refinement.**
> RLIE uses calibrated probabilities to identify representative hard examples, then feeds them back together with the current rule bank to the LLM for the next round of rule generation. Thus, the aggregator is not merely a final readout; it also affects subsequent rule learning.
>
> **Direct quantitative comparison.**
> To quantify this difference, we directly transplant Zhong et al.’s  pipeline to our tasks and compare it with **RLIE (linear-only)** under the same setting, using the same **DeepSeek-V3.2** backbone and averaging over **3 runs**.
>
> | Method | Reviews | Dreaddit | Headlines | Citations | LLM Detect | Retweets |
> |---|---:|---:|---:|---:|---:|---:|
> | Zhong et al | 66.2 / 65.9 | 75.4 / 75.0 | 48.2 / 44.1 | 58.1 / 54.7 | 84.9 / 84.6 | 50.0 / 52.1 |
> | **RLIE (linear-only)** | **70.9 / 70.7** | **82.3 / 82.3** | **67.0 / 67.0** | **64.6 / 63.0** | **90.7 / 90.7** | **65.7 / 65.6** |
>
> RLIE is stronger on all 6 tasks. In particular, in terms of F1, **Headlines** improves by **+22.9** and **Retweets** by **+13.5**. The gains come from **predictive rules with abstention**, **joint optimization of the full rule set**, and **probability-guided closed-loop refinement**.

---

> > ### Author Rebuttal · Reviewer_oqmB · 2026-04-03
> >
> > I adjusted my score to accept based on your response.  Thanks for your comparison.

---

> > > ### Author Response · Authors · 2026-04-03
> > >
> > > Thank you very much for your thoughtful feedback. We sincerely appreciate your careful reading of our rebuttal, and are grateful that you found many of our clarifications helpful and increased your score.

---

### Official Review · Reviewer_zhQp · 2026-03-12

**Soundness:** 3
**Presentation:** 3
**Significance:** 2
**Originality:** 2
**Overall Recommendation:** 3
**Confidence:** 4

**Summary:**

This paper studies how to learn interpretable natural-language rule sets for text classification with LLMs. The proposed RLIE pipeline asks an LLM to generate candidate rules, uses an LLM judge to apply each rule to each instance with ternary outputs in {-1, 0, +1}, fits an elastic-net logistic regression model on these rule-level outputs, and iteratively refines the rule set using hard examples. The paper evaluates this framework on six text classification tasks and multiple LLM backbones. Beyond the main performance comparison, it also studies different inference protocols and argues that LLMs are useful for rule induction and local rule application, while an explicit statistical combiner is more reliable than prompt-based aggregation for final prediction.

**Compliance With Llm Reviewing Policy:**

Affirmed.

**Final Justification:**

Thank you for the rebuttal. The reply mainly strengthens the framing, not the originality case. Addressing several prior shortcomings supports integration; an adaptive rule bank still resembles iterative candidate refinement; explicit aggregation remains a fairly standard sparse probabilistic combiner; probability-guided updating still looks like a familiar feedback loop; and the added evaluations support effectiveness rather than novelty. Overall, this reads more like a well-structured closed-loop system design than a substantially new methodological mechanism. So, I keep my original score.

**Key Questions For Authors:**

1. My main reservation is originality. The method currently reads to me as LLM-based rule generation/application followed by a standard sparse linear combiner. If the authors can clearly explain what is genuinely new in the formulation, rather than only in the system combination, I would be open to revising my originality score upward.

2. I would also appreciate a sharper formal characterization of RLIE. At present, I understand it as a two-level model with an LLM-defined discrete intermediate representation and a trainable linear readout. If the authors believe this characterization is incomplete, please clarify what the more faithful abstraction is. A clearer formulation could change my view of the paper’s conceptual contribution.

3. Finally, I encourage the authors to explicitly state what they see as the single most important contribution of the paper: a new method, a new perspective on neuro-symbolic decomposition, or a strong empirical finding about the limitations of prompt-based aggregation. Depending on that clarification, I may revise my assessment of significance and originality.

**Limitations:**

Yes

**Strengths And Weaknesses:**

Strengths:

- The paper addresses a relevant problem: how to obtain interpretable natural-language rules from LLMs while avoiding brittle prompt-only aggregation. The proposed division of labor between rule induction and rule aggregation is intuitive and practically meaningful.
- The method is technically straightforward and mostly sound. Using an elastic-net logistic combiner over ternary rule outputs is a reasonable design choice for sparse and interpretable aggregation, and the comparison of inference protocols (E1-E4) is one of the most useful aspects of the paper.
- The empirical study is reasonably broad: six tasks, multiple LLM backbones, cost analysis, calibration analysis, and qualitative case studies. The central empirical takeaway—that explicit aggregation is generally more stable than prompt-based use of rules and weights—is interesting and potentially useful to the community.
- The paper is generally clearly written. The high-level pipeline is easy to follow, and the figures/case studies help communicate the intended behavior of the method.

Weaknesses:

- The methodological novelty is limited. Once rule-level outputs are available, learning sparse linear weights is a very standard and arguably the simplest downstream design. Hard-example-driven refinement is also a familiar idea. As a result, the main contribution is more a careful system combination and empirical validation than a substantially new method.
- Some soundness and clarity issues remain. The paper does not sufficiently isolate how much of the gain comes from (i) the explicit combiner, (ii) iterative refinement, and (iii) the quality/stability of the LLM judge that produces rule-level outputs. Additional ablations would strengthen the claims.
- There is an inconsistency regarding model selection: the main text states that logistic-regression hyperparameters are selected on the validation set, while the appendix states that they are chosen via 5-fold CV on the training set. This should be clarified precisely, since it affects the interpretation of the experimental protocol.
- The practical significance is moderate rather than high. Although the final predictor is linear, inference still appears to require LLM-based per-rule judgments for each test instance, which may limit scalability and deployment practicality. The comparison to simpler alternatives such as embedding-based classifiers or linear probes over frozen LLM representations is also missing.
- The related-work positioning could be sharper. The paper differentiates itself from prior LLM hypothesis-generation and prompt-based rule-use methods, but it under-emphasizes how close the final aggregation step is to classical rule ensembles / sparse linear models over interpretable features. This matters when judging originality.

Overall assessment by dimension:

- Soundness: mostly good, but several important ablations and protocol clarifications are still needed.
- Presentation: good overall, with a clear narrative, though the novelty claims should be stated more carefully and the experimental protocol should be clarified.
- Significance: moderate. The empirical message is useful, but the practical and conceptual advance is somewhat limited by the simplicity of the downstream model and the remaining dependence on LLM-based rule application.
- Originality: fair. The paper offers a reasonable combination of existing ideas and a useful empirical perspective, but the core method itself is incremental.

---

> ### Author Rebuttal · Authors · 2026-03-30
>
> We thank the reviewer for the careful feedback and recognition of the **practical value** of our **rule induction/aggregation decomposition** and **inference-protocol comparisons**. We address your concerns point by point below.
>
> > **Weakness 1:** *The novelty looks more like integration than a new method.*
>
> RLIE’s contribution lies in its **closed-loop formulation**: the rule set is adaptive, explicit probabilistic aggregation is part of learning rather than only a final readout, and learned probabilities feed back to update the rule bank.
>
> For more details, please see **Questions 1–3 below.**
>
> > **Weakness 2:** *The roles of the combiner, judge, and refinement are not cleanly isolated.*
>
> **Combiner.** Replacing the LR aggregator with **unweighted majority voting lowers validation ACC on all 6 tasks**, with an average drop of **0.0502**. Full results are reported in [Table 3](https://anonymous.4open.science/r/supp-tables-5C58/RLIE_rebuttal_external_tables.md).
>
> **Judge.** Across all 6 tasks, pairwise **non-abstain agreement** between different judge backbones remains at **0.7964–0.9204, demonstrating strong consistency between different LLM backends.** For more details, please see **Reviewer RM1Z, Question 1.**
>
> **Refinement.** We compare the initial rule set (Round 0) against the best validation performance after iterative refinement. On all 6 tasks, **the best refined results improve F1 by 0.055 on average over the initial version**, showing the value of refinement. For more details, please see **Reviewer RM1Z, Weakness 3, and [Table 4](https://anonymous.4open.science/r/supp-tables-5C58/RLIE_rebuttal_external_tables.md).**
>
> > **Weakness 3:** *The description of the validation-selection protocol is inconsistent.*
>
> Thanks for pointing out this typo. In the actual implementation, logistic-regression hyperparameters are selected by **5-fold cross-validation on the training set**. The validation split is used for **best-round selection and early stopping**. So this is a wording error, not data leakage. We will correct this wording in the revision.
>
> > **Weakness 4:** *The practical significance is limited because inference still requires per-rule LLM judging.*
>
> This is a deliberate **cost–interpretability tradeoff**: RLIE chooses to keep an explicit natural-language rule interface rather than compressing everything into a black-box classifier, thereby retaining interpretability and controllability.
>
> **In practice,** most cost is concentrated in offline **rule proposal / refinement**; afterwards, a cheaper model can be used for online rule judging, possibly with lightweight adaptation.
>
> > **Weakness 5:** *Comparisons to simpler alternatives such as embedding-based classifiers or linear probes are missing.*
>
> We additionally fine-tuned a Qwen3-8B classifier baseline with LoRA.
>
> **RLIE improves average test F1 by 0.062 over all 6 tasks**, while also producing **reusable explicit rules**. Full per-task comparisons are in [Table 5](https://anonymous.4open.science/r/supp-tables-5C58/RLIE_rebuttal_external_tables.md).
>
> > **Weakness 6:** *The related-work positioning with respect to sparse linear / rule-based models is still unclear.*
>
> RLIE differs not by using a new aggregator, but at the **system level**: it learns **predictive rules**, rules can **abstain**, and the learned **probabilities feed back** into hard-example selection, pruning, and refinement under explicit capacity control.
>
> In **Weakness 2** we compare against **non-probabilistic majority-vote aggregation**, and in **Reviewer oqmB, Question 1** we compare directly against **Zhong et al.** Both show consistent advantages.
>
> > **Question 1:** *What is genuinely new in RLIE?*
>
> RLIE’s novelty lies in a **closed-loop formulation for adaptive natural-language predictive rule learning**, where learned probabilities guide hard-example selection and rule-bank updates under explicit rule-set control.
>
> For more details, please see **Reviewer eooD, Weakness 1.**
>
> > **Question 2:** *What is a more accurate formal characterization of RLIE?*
>
> **RLIE is a capacity-controlled, probability-guided, closed-loop predictive neuro-symbolic system.**
>
> Its neural part proposes and applies **natural-language rules**; its **symbolic-probabilistic** part explicitly aggregates those rules and feeds the resulting probabilities back to determine which samples are hard, which rules should be kept, and how the rule bank should be refined.
>
> Therefore, RLIE is **not** a static two-stage pipeline with a final linear readout. More accurately, **the rule bank, its effective size, and its probabilistic readout co-evolve during learning.**
>
> > **Question 3:** *What is the single most important contribution of this paper?*
>
> RLIE proposes a predictive **neuro-symbolic system** for **LLM-based rule learning and inference**, unifying adaptive rule-set control and explicit probabilistic rule reasoning in the same **closed loop** while retaining both **interpretability and performance**.

---

> > ### Author Rebuttal · Reviewer_zhQp · 2026-04-03
> >
> > Thank you for the thoughtful rebuttal. I found the added analyses helpful, especially the new evidence on judge robustness, the role of the probabilistic combiner, the refinement/capacity ablations, and the clarification of the train/validation protocol. These additions improve my confidence in the technical soundness of the work and make the paper’s intended framing clearer.
> >
> > However, after considering the rebuttal, I will keep my original score. My main concern is not with the empirical validity of the framework, which is now better supported, but with the overall level of novelty. I still see the paper’s contribution as lying primarily in a well-executed system formulation and empirical insight, rather than in a sufficiently strong methodological advance to change my overall recommendation.

---

> > > ### Author Response · Authors · 2026-04-03
> > >
> > > We thank the reviewer for the further comments and the opportunity to clarify our contribution. Our point is not that RLIE introduces a new standalone combiner, but that it addresses several concrete shortcomings of **LLM-based natural-language rule learning and inference**: prior methods often refine one hypothesis at a time or maintain a loose candidate pool [1], leave multi-rule aggregation implicit inside prompting [2], or use explicit statistical models without a probability-guided closed loop for updating a predictive rule bank [3,4].
> > >
> > > 1) **RLIE treats the rule set itself as a learning object**.
> > >
> > > Rather than assuming a single rule or a fixed candidate bank, it learns which rules to keep and how many to keep, so the rule bank can be expanded, pruned, and refined [1].
> > >
> > > 2) **RLIE performs explicit probabilistic aggregation over natural-language rules**.
> > >
> > > This differs from prompting-based rule use, where multi-rule aggregation remains implicit [2], and from natural-language parameter models whose objects are closer to predicates/parameters than to an adaptively maintained predictive rule bank [3].
> > >
> > > 3) **RLIE uses probabilities not only for final prediction, but also for rule-bank updating**.
> > >
> > > Learned probabilities feed back into hard-example selection, pruning, and subsequent refinement. Related work has studied probabilistic reasoning over natural-language hypotheses [4], but not this kind of predictive, capacity-controlled, multi-rule bank learning with probability-guided updates.
> > >
> > > 4) **RLIE is a paradigm-level decomposition** rather than a loose engineering combination.
> > >
> > > As in interpretable intermediate-representation work [5,6], RLIE separates local rule induction from global prediction; however, here the intermediate layer is an evolving **natural-language rule bank**. The LLM proposes and applies local rules, the probabilistic model performs global aggregation and calibration, and the resulting probabilities drive further updates of the rule bank. Our added evaluations support precisely this claim: this decomposition yields a more stable, analyzable, and effective learning-and-inference process, not an arbitrary combination.
> > >
> > > **We believe RLIE contributes a useful direction for rule learning in the LLM era**. It suggests that natural-language rules should not remain auxiliary prompt descriptions, but should be treated as explicit, compositional, and updatable reasoning objects. In particular, **probabilistic evaluation and probability-guided updating** may be useful design principles for future work in interpretable LLM-based rule learning and reasoning.
> > >
> > > **References**
> > >
> > > [1] Yangqiaoyu Zhou, Haokun Liu, Tejes Srivastava, Hongyuan Mei, and Chenhao Tan. 2024. **Hypothesis Generation with Large Language Models**. In *Proceedings of the 1st Workshop on NLP for Science (NLP4Science)*, pages 117–139, Miami, FL, USA. Association for Computational Linguistics.
> > >
> > > [2] Sergio Servantez, Joe Barrow, Kristian Hammond, and Rajiv Jain. 2024. **Chain of Logic: Rule-Based Reasoning with Large Language Models**. In *Findings of the Association for Computational Linguistics: ACL 2024*, pages 2721–2733, Bangkok, Thailand. Association for Computational Linguistics.
> > >
> > > [3] Ruiqi Zhong, Heng Wang, Dan Klein, and Jacob Steinhardt. 2024. **Explaining Datasets in Words: Statistical Models with Natural Language Parameters**. In *Advances in Neural Information Processing Systems 37 (NeurIPS 2024)*.
> > >
> > > [4] Wasu Top Piriyakulkij, Cassidy Langenfeld, Tuan Anh Le, and Kevin Ellis. 2024. **Doing Experiments and Revising Rules with Natural Language and Probabilistic Reasoning**. In *Advances in Neural Information Processing Systems 37 (NeurIPS 2024)*.
> > >
> > > [5] Pang Wei Koh, Thao Nguyen, Yew Siang Tang, Stephen Mussmann, Emma Pierson, Been Kim, and Percy Liang. 2020. **Concept Bottleneck Models**. In *Proceedings of the 37th International Conference on Machine Learning (ICML 2020)*, *Proceedings of Machine Learning Research*, volume 119, pages 5338–5348. PMLR.
> > >
> > > [6] Yue Yang, Artemis Panagopoulou, Shenghao Zhou, Daniel Jin, Chris Callison-Burch, and Mark Yatskar. 2023. **Language in a Bottle: Language Model Guided Concept Bottlenecks for Interpretable Image Classification**. In *Proceedings of the IEEE/CVF Conference on Computer Vision and Pattern Recognition (CVPR 2023)*, pages 19187–19197.

---

### Official Review · Reviewer_eooD · 2026-03-13

**Soundness:** 3
**Presentation:** 3
**Significance:** 3
**Originality:** 3
**Overall Recommendation:** 4
**Confidence:** 2

**Summary:**

This paper proposes RLIE, a unified framework for learning compact, interpretable natural-language rule sets by integrating LLM-based hypothesis generation with a calibrated probabilistic combiner and error-driven refinement. The authors verify the effectiveness of RLIE on six datasets, showing its high performance.

**Compliance With Llm Reviewing Policy:**

Affirmed.

**Final Justification:**

Rebuttal addressed my concerns, and I kept the score.

**Key Questions For Authors:**

1. What is the key mechanism of the proposed method, beyond the simple prompt engineering? The authors should emphasize such part throughout the whole paper.

2. Can authors provide the results using commercial-level LLMs like Gemini or GPT?

3. Can authors provide additional ablation studies, especially regarding sensitivity of hyperparameters? For example, how much performance drops if we use fewer than 5 rules?

**Limitations:**

yes

**Strengths And Weaknesses:**

**Strength**

1. Method is quite simple while it shows high performance.

2. I think some components like obtaining a weight for each rule by Logistic Regression and Refining rule sets iteratively is quite novel.

**Weakness**

1. Method is only done by prompting LLMs. It can be seen as just simple prompt engineering.

---

> ### Author Rebuttal · Authors · 2026-03-30
>
> We thank the reviewer for recognizing the value of framework and address concerns below.
>
> > **Weakness 1:** *The method still relies on prompting and therefore looks like simple prompt engineering.*
>
> **At the method level,** RLIE uses prompting to propose candidate rules and obtain rule-level judgments, but it does not directly output the final prediction.
>
> **At the system level,** the real contribution is a **closed-loop predictive rule-learning framework** that combines semantic rule induction, explicit probabilistic aggregation, adaptive rule-set control, and probability-guided refinement.
>
> RLIE **gains from a deliberate division between local semantic judgments and global probabilistic decisions**, which enables explicit refinement and rule-bank control rather than simple prompting.
>
> For more details, please see **Question 1** below and **Reviewer zhQp, Questions 1-3.**
>
> > **Question 1:** *Beyond simple prompt engineering, what is the key mechanism of the method?*
>
> The key mechanism and contribution of RLIE is a **closed-loop division of labor between local semantic judging and global probabilistic learning.**
>
> **During rule learning:**
>
> **(1) The aggregator is inside the learning loop.**
> RLIE first learns a probabilistic aggregator over rule activations, then uses its calibrated probabilities to identify hard examples and drive the next round of rule proposals.
>
> **(2) Rule complexity is explicitly controlled.**
> RLIE does not rely on a rule bank with a fixed predefined size. Through regularization, reweighting, and pruning, it maintains a compact and adaptive rule bank.
>
> **During inference:**
>
> **(1) The LLM performs only local judging, not final classification.**
> It decides whether each rule applies to the current sample, can abstain when a rule has no substantive applicability, and provides only **local semantic evidence.**
>
> **(2) The final decision is explicit and global.**
> The probabilistic aggregator more stably combines all rule activations, rather than relying on in-context rule aggregation.
>
> Overall, this neuro-symbolic decomposition provides a more **interpretable and flexible framework** for predictive rule learning, and empirically **improves the stability and performance** of rule-based reasoning.
>
> **For standardized novelty phrasing, see Reviewer zhQp, Questions 1-3.**
>
> > **Question 2:** *Can you provide results using commercial-level LLMs such as Gemini or GPT?*
>
> Under the **same protocol**, we additionally tested **RLIE and all baselines** with **GPT-5.4 (no-thinking)** while keeping the same **linear-only** setting.
>
> Empirically, **RLIE is still the best on all 6 tasks.** In terms of **test F1**, it **exceeds** the second-best baseline by about **5.7 points on average**. Full per-method results are provided in [Table 1](https://anonymous.4open.science/r/supp-tables-5C58/RLIE_rebuttal_external_tables.md).
>
> This suggests that RLIE’s advantage does not depend on any particular backbone. At the same time, it remains consistently stable on weaker models, further supporting the method’s **generality, robustness, and backbone-independence**.
>
> > **Question 3:** *Can you provide additional ablation studies, especially on hyperparameter sensitivity?*
>
> On the **headline** dataset, we directly **ablated the maximum rule-bank size H**, testing rule budgets of **2 / 3 / 5 / 10 / 20 / 30**, and under the same DeepSeek V3.2 protocol as the main experiments we jointly report both **performance** and **total token usage**. The results show that **our method is relatively robust** to H. Full results are provided in [Table 2](https://anonymous.4open.science/r/supp-tables-5C58/RLIE_rebuttal_external_tables.md).
>
> **Some details are as follows:**
>
> **(1) Very small rule banks cannot fully realize RLIE’s advantage.**
> When only **2-3 rules** are allowed, the average F1 over all tasks is only **0.636-0.644**, whereas with **5 rules** it reaches **0.682**. The reason is that an overly small rule bank cannot fully exploit multi-rule complementarity to capture the complexity of real task patterns.
>
> **(2) Once capacity reaches a moderate range, performance becomes stable.**
> With **5 / 10 / 20 / 30** rules, F1 stays within a relatively narrow range of **0.656-0.682**, showing that RLIE does **not depend** on an extremely large rule bank. This also suggests that real tasks usually do not require an extremely complicated rule set. Under a moderate capacity budget, RLIE can already **adaptively learn sufficiently effective rules** and stably realize the advantage of multi-rule compositional modeling.
>
> **(3) Cost is reported jointly and remains controllable overall.**
> Across these settings, total token usage is **5.72M-22.45M**. Cost is not strictly monotonic, because smaller budgets may trigger extra refinement; for example, **3 rules (5.72M)** is actually cheaper than **2 rules (7.29M)**. When the rule count further increases to 20 and 30, total cost rises overall, but the **increase remains controllable**.

---

> > ### Author Rebuttal · Reviewer_eooD · 2026-04-03
> >
> > Thanks for the rebuttal, and since I was initially positive to this paper, I remain the score.

---

> > > ### Author Response · Authors · 2026-04-03
> > >
> > > Thank you very much for your thoughtful feedback. We sincerely appreciate that you found our rebuttal helpful and that we were able to address your concerns.

---

### Decision · Program_Chairs · 2026-04-30

**Decision:**

Accept (regular)

**Comment:**

The paper presents a simple and practical idea: to generate interpretable decision rules, ask an LLM to generate rules and then combine them using standard regression techniques. The extensive empirical results support the efficacy of the method in finding interpretable and high accuracy solutions.

Overall, I think the paper introduces an effective, practical idea and supports it with strong evidence. There are still some limitations--e.g., LLM reliance in many stages--- that can be improved. During the rebuttal, authors also compared their work to a recent related work. I suggest to include this comparison in the main paper.